

# Experimental investigation of wind turbine controllers for the Hybrid-Lambda Rotor

Daniel Ribnitzky[1,2], Vlaho Petrović[1,2], and Martin Kühn[1,2]

[1]Carl von Ossietzky Universität Oldenburg, School of Mathematics and Science, Institute of Physics
[2]ForWind - Center for Wind Energy Research, Küpkersweg 70, 26129 Oldenburg, Germany

**Correspondence:** Daniel Ribnitzky (daniel.ribnitzky@uol.de)

**Abstract.** The continuous growth in rotor diameter of offshore wind turbines must be accompanied by advanced control strategies that master the trade off between limiting extreme loads and maximizing power output, fostering a lightweight and cost-effective blade design. This is addressed by the Hybrid-Lambda Rotor design and control methodology which realizes two operating modes by following two different tip speed ratios (TSRs) below rated power with an overarching load constraint.

Contrary to conventional wind turbine controllers, this leads to a wide range of wind speeds where the torque and pitch controllers are active simultaneously. The objective of this paper is to develop and apply such control strategies on the MoWiTO 1.8 model wind turbine and to experimentally validate them under turbulent reproducible inflow conditions in the wind tunnel using an active grid. The results are examined regarding extreme loads, power production, fatigue loads and pitch actuation. Further, we discuss the scaling of the controller characteristics and inflow test cases according to the model turbine scaling.

Different versions of the pitch controller are introduced. First, a baseline controller with a model-based wind speed estimator which performed well in tracking the different TSRs. Second, a load feedback controller that overcame model uncertainties and performed well in setting the mean value of the loads. And third, an inflow feed-forward controller which was able to reduce load overshoots in gust events. With the results presented here, we make the next step in the experimental validation of the control methodology, which unlocks the full potential of aerodynamic efficiency and ensures the structural integrity of the

Hybrid-Lambda Rotor.

## 1 Introduction

The size of future wind turbine rotors is continuously increasing and the current trend of growth in rotor diameter does not seem to saturate. Rotors with a low-specific rating can capture more energy in light winds which is beneficial for a reliable and more continuous electricity supply and it will increase the value of wind energy. However, large rotors can only be designed

economically feasible if extreme loads are limited, fostering a lightweight and cost-effective blade design. This emphasises the necessity of advanced control algorithms which limit the extreme loads whenever needed but maximize the power output whenever possible.



This problem is addressed by the Hybrid-Lambda rotor design methodology. It describes how the rotor diameter can be increased while maintaining the rated power and the extreme loads of a respective reference turbine. The optimization process can be described as the maximization of the power output over the entire partial load range while constraining the flapwise blade root bending moment. The blade design methodology includes a design for two tip speed ratios (TSRs) and two operating modes below rated power which is closely coupled to the controller design. Previous studies have successfully shown the benefits of the concept compared to conventional upscaling approaches. The Hybrid-Lambda concept was first applied on a 15 MW offshore wind turbine with a specific rating of $180 \, \mathrm{W \, m^{-2}}$ and a diameter of 326 m. In simulation environments, the concept was investigated using steady state operating points and transient load cases including a preliminary controller (Ribnitzky et al., 2024). Further, the concept was scaled to wind tunnel size and verified for steady state operating points (Ribnitzky et al., 2025). A detailed explanation of the controller design and an experimental investigation of the transient behaviour of the Hybrid-Lambda controller are still missing. With this paper, we aim to fill this gap.

Limiting extreme operating loads is often addressed by conventional peak-shaving. I.e. the minimum pitch angle is calculated as a function of the wind speed in order to limit the load to a constant value. The controller then uses a wind speed estimator and chooses the pitch angle from a look-up table (LUT), as described among others by Abbas et al. (2022), where the torque controller is not adjusted for peak-shaving purposes. Pusch et al. (2024) and Lazzerini et al. (2025) introduced frameworks to determine optimal steady-state operating points (pitch angle and rotor speed) for wind turbine control over the entire operating range of wind speeds, by solving optimization problems constrained by maximal loads or rotor speed values. Lowering the mean value of a certain load, as done for conventional peak-shaving, will reduce the peak fluctuations, too, as shown by Bottasso et al. (2014). But, it does not yet ensure that the peak fluctuations will not cross a certain limit. This can be realized by envelope protection algorithms as presented by Petrović and Bottasso (2017). Other approaches focus on feed-forward controllers that enable a preview on the inflow and allow the turbine to act in advance rather than reacting on load overshoots. Guo et al. (2023) and Fu et al. (2023) described how a feed-forward controller can be set up using preview measurements acquired with a lidar. Schlipf et al. (2013) also used model predictive control for this purpose. The advanced control methods improved the turbine response with a reduction in rotor speed and power variations, as well as a reduction in extreme and fatigue loads. Sinner et al. (2022) presented the first physical test of model predictive control for blade pitch control of a scaled wind turbine which effectively reduced rotor speed variations above rated wind speed. They used the Model Wind Turbine Oldenburg (MoWiTO 1.8) with 1.8 m diameter (Berger et al., 2018) in the $3 \times 3 \, \mathrm{m}$ wind tunnel in Oldenburg (Kröger et al., 2018) and tested the controller under reproducible turbulent inflow conditions generated with an active grid (Neuhaus et al., 2021). This set-up is also used in the study presented here. Bottasso and Campagnolo (2021) explained fundamentals on the scaling of wind turbines and wind tunnel testing as well as the design of the controllers and actuators for scaled turbines. Experimental validation of innovative control algorithms bridges the gap between simulation-based studies and field testing. The advantage of wind tunnel tests is the ability to test under tailored and reproducible inflow conditions. However, when using model turbines, the effects of scaling must be carefully considered. Wind tunnel testing adds important value compared





to simulation-based studies while avoiding administrative hurdles of full-scale tests.

The objective of this paper is to develop control methodologies for very large wind turbines with a load constraint that follow the Hybrid-Lambda operation strategy. This includes the realization of two operating modes with corresponding TSR values below rated power. The control methods are applied on a 1.8 m diameter model wind turbine and the required scaling conditions are derived. Four controller versions are tested in the wind tunnel under reproducible turbulent inflow conditions. The experimental testing will provide answers to the following research aspects. We address how the transition between the

operating modes can be accomplished. We investigate how we can achieve TSR tracking while the pitch controller is simultaneously active to constrain the loads. We explore different methods on how to constrain the extreme operating loads and how to deal with model uncertainties. Finally, we evaluate the controller versions considering extreme loads, fatigue, pitch actuation and power output.

## 2    Methodology

In this section, we first explain the Hybrid-Lambda control methodology. The transferability of the experimental results from wind tunnel scale to the full-scale turbine is demonstrated in Sect. 2.1.1. We then address the torque controller in Sect. 2.1.2 and we outline four different versions of the pitch controller in Sect. 2.1.3. The experimental set-up in the wind tunnel is described in Sect. 2.2, followed by an explanation of the test cases in Sect. 2.3 and the method to calculate damage equivalent loads and pitch actuation in Sect. 2.4.

### 2.1    The Hybrid-Lambda control methodology

In this section we describe the control schedule for the Hybrid-Lambda Rotor, as depicted in Fig. 1 for the scaled wind tunnel model. We provide more information on the full-scale blade design, which is derived for considerably higher TSRs, in our previous publication (Ribnitzky et al., 2024) and on the aerodynamic scaling to wind tunnel size in (Ribnitzky et al., 2025). A major objective of the Hybrid-Lambda rotor design and control methodology is to achieve better power coefficients whenever

loads need to be constrained. Therefore, the rotor is designed for two operating modes below rated power. In the light-wind (LW) mode, the rotor is operated at a high TSR (7.5 in case of the model wind turbine) and with the optimum pitch angle (fine pitch is -0.8°). Since the outer 30% of the blade is designed for a TSR of 7.5, the light-wind mode assures the maximum power coefficient. At a certain wind speed, the maximum loads are reached, in this case the maximum flapwise blade root bending moment (RBM) of 7.3 Nm. This wind speed is called $u_{ts} = 6.3 \, \mathrm{m \, s^{-1}}$, marking the start of the transition to the strong-wind

mode for higher wind speeds. In the transition region, the rotational speed is kept constant at $\omega_{trans} = 500 \, \mathrm{rpm}$ in order to reduce the TSR and the pitch is increased to constrain the flapwise RBM. Once the lower TSR of 6 is reached, the transition ends at the wind speed $u_{te} = 7.9 \, \mathrm{m \, s^{-1}}$. For higher wind speeds, the rotor operates in the strong-wind mode, following the lower TSR of 6 and the pitch is increased accordingly to limit the loads. Since the inner 70% of the blade is designed for a TSR of 6, the inner part of the blade is operated in its design point in the strong-wind (SW) mode. The loads in the outer part are relieved





and the lever arm of the resulting bending forces is reduced. This is the reason why better power coefficients can be achieved compared to conventional peak-shaving strategies where the blade is conventionally optimised for one operating point and the load constraint is achieved solely by pitching to feather. These advantages are thoroughly analysed in a previous publication (Ribnitzky et al., 2024). In contrast, this study focuses on how this control schedule can be realised by a pitch and torque controller.


The Hybrid-Lambda control methodology introduces several additional features that are not present for conventional wind turbine controllers. First, the rotational speed needs to be kept constant in a defined wind speed range to incorporate the transition between the operating modes below rated wind speed. Second, the pitch angle needs to be set below rated wind speed in order to constrain the maximum loads in the transition region (constant rotational speed) and in strong-wind mode (variable rotational speed). And third, the torque controller is supposed to track the lower TSR in the strong-wind mode, but the pitch angle will vary with respect to the wind speed. Therefore the torque coefficient $c_q$ will not be constant, and the proportionality $k$ in the $k\omega^2$-law of the torque controller is not constant either. The generator torque $M_g$ is calculated as expressed in Eq. 1. Here, $R$ is the rotor radius, $\rho$ is the air density, $\lambda$ is the tip speed ratio, $\beta_{pitch}$ is the pitch angle and $\omega$ is the rotational speed. Overall, the pitch and torque controllers will be active simultaneously over a wide range of wind speeds, which is usually not the case for conventional wind turbine controllers.

$$M_g = \frac{\pi R^5 \rho c_q(\lambda, \beta_{pitch})}{2 \lambda^2} \omega^2 = k\omega^2 \qquad (1)$$

Compared to the control schedule presented by Ribnitzky et al. (2025), the maximum rotational speed is reduced to 577 rpm to keep a sufficient margin to the hardware constraint of 600 rpm. Note, that the pitch is not adjusted for very low wind speeds when the turbine operates at minimum rotational speed. This could further increase the power output close to cut-in.



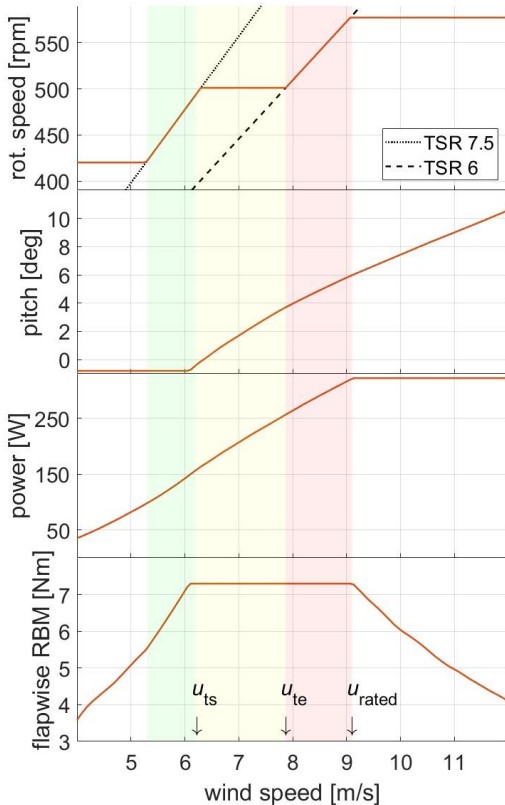

**Figure 1.** Control schedule for the Hybrid-Lambda model turbine, derived from steady-state measurement data. Background colours indicate the operating mode: green, light-wind mode; yellow, transition; red, strong-wind mode.

### 2.1.1 Scaling considerations

When testing wind turbine applications in the wind tunnel, we need to consider scaling effects. It has to be ensured that the experimental set-up represents the respective full-scale application reasonably well. However, not all physical characteristics can be matched simultaneously and compromises must be found. The scaling of the rotor aerodynamics was successfully

demonstrated by Ribnitzky et al. (2025). In this section, we address the scaling of parameters, that are relevant for controller evaluations, which are summarized in Table 1.

We introduce scaling factors $n$ which are defined by dividing the parameter of the model turbine (subscript m) by the parameter of the full-scale turbine (subscript f). The geometric length scaling factor $n_l$ is defined by the ratio of the rotor

diameters $D$.

$$n_l = \frac{D_m}{D_f} = \frac{1}{181} \qquad (2)$$



Scaling the rotational inertia of the entire rotor is not possible for the given geometric scaling ratio. In theory, the rotational inertia would scale with $n_l^5$. But due to structural constraints, the scaled blades can not be manufactured light enough to fulfil this scaling criterion. Consequently, the rotational inertia of the model turbine is about six times larger than required from the scaling law.

The actuation system is subjected to the time scaling $n_t$, which is derived by the ratio of rotational speeds in the transition region.

$$n_t = \frac{t_m}{t_f} = \frac{\omega_{trans,f}}{\omega_{trans,m}} = \frac{1}{114} \tag{3}$$

For the full-scale 15 MW Hybrid-Lambda turbine, Ribnitzky et al. (2024) assumed a maximum pitch rate of $3\,°/s$. The hardware constraint for the model turbine is at $85\,°/s$. To match the exact time scaling, an actuation rate of $342\,°/s$ would be necessary. This means, the pitch system of the model turbine is about four times slower than the full-scale equivalent.

The maximum aerodynamic torque would scale with $n_l^3$, assuming a constant TSR, identical torque coefficients and identical inflow wind speeds. In fact, these assumptions do not hold for the given rotor scaling, but we provide further clarification in the appendix A, and we explain why the maximum rotor torque is still similar as to this simplified scaling. Assuming that the rotor torque scales with $n_l^3$, the maximum torque of the model turbine is about 14% higher than derived from the scaling law. For the given model turbine, the generator torque rate is ideally scaled with $n_l^3/n_t$ and constrained through the control software. Consequently, the actuation time (e.g. time from zero to maximum torque) is 14% slower for the model turbine compared to the full-scale equivalent. Maximal pitch and torque rates represent theoretical limits and it is worth noting that the controller rarely operates at the said limits. Further details are provided in the results sections of the respective test cases.

As we will further discuss in Sect. 4, also the inflow is subject to scaling constraints. The inflow can not be adjusted arbitrarily fast, due to the inertia of the air mass and due to hardware constraints of the actuation systems in the wind tunnel. As an example, the gust event analysed in Sect. 3.2 lasts about eight times longer than the full-scale equivalent.

To further compare the system behaviour, we can calculate the time constants in the equation of motion for the two rotors. Those can be modelled as a non-linear first-order system.

$$I \frac{d\omega}{dt} = M_a(\omega, u, \beta_{pitch}) - M_g(\omega) \tag{4}$$

Here, we neglected the friction losses in the drive train for simplicity. $M_a$ and $M_g$ are the aerodynamic and generator torque, respectively. If we linearize this system around an initial rotational speed $\omega_0$ with a small perturbation of $\delta\omega$ we derive:

$$I \frac{d}{dt} \delta\omega \approx \left( \frac{dM_a}{d\omega} - \frac{dM_g}{d\omega} \right) \delta\omega \tag{5}$$

This is a first-order linear ordinary differential equation with the time constant $T$

$$T = \frac{I}{\frac{dM_g}{d\omega} - \frac{dM_a}{d\omega}} \tag{6}$$





and the solution

$$\omega(t) = \omega_0 + c_1 \cdot e^{-t/T}, \ \text{ for } \ t > 0. \tag{7}$$

The derivative of the aerodynamic torque for the perturbation in the rotational speed under constant wind speed can be calculated as follows:

$$M_a = \frac{1}{2}\rho\pi R^3 u^2 c_q(\lambda,\beta) \tag{8}$$

$$\lambda = \frac{\omega R}{u} \tag{9}$$

$$\frac{dM_a}{d\omega} = \frac{dM_a}{d\lambda} \cdot \frac{d\lambda}{d\omega} = \frac{1}{2}\rho\pi R^4 u \frac{dc_q(\lambda,\beta)}{d\lambda} \tag{10}$$

We calculate the time constants for the light-wind and strong-wind operating modes (for the wind speeds $u_{ts}$ and $u_{te}$) with the Eq. 6 and the derivative of the aerodynamic torque from Eq. 10. The required data can be found in Table A2. The derivative of the generator torque can be derived from the LUT which is used by the controller (mapping generator torque over rotational speed). Here, we use numerical backwards differentiation for $u_{ts}$ and numerical forwards differentiation for $u_{te}$ to avoid the ambiguity at $\omega_{trans}$ as further explained in the next sub-section. We compare the resulting time constants for the full-scale and the model turbine in Table 1.

Considering the time scaling factor $n_t = 1/114$, the model turbine reacts about two times slower in the light-wind mode and about 1.5 times slower in the strong-wind mode to a perturbation in the inflow, compared to an ideally scaled representation of the full-scale turbine.

In conclusion, the model turbine represents a very large and heavy rotor with a high rotational inertia. Considering the change in aerodynamic torque with respect to a change in the rotational speed (or in the inflow), also the time constant of the system response is about two times higher for the model turbine. Further, the actuation hardware represents a slow system for the full-scale application. Consequently, these scaling limitations represent a conservative approach. I.e., despite the tremendous size (15 MW, 326 m diameter) of the full-scale system, it will most likely perform equally well (if not better) in terms of controller actuations and system response times, compared to the model turbine. Only the slower representation of gust events would lead to a non-conservative scaling. However, the effects of slower inflow variations are partly compensated by the slower turbine response time and slower actuation rates.



**Table 1.** Scaling of controller parameters for the Hybrid-Lambda Rotor.

| Parameter | Symbol | Model scale (subscript m) | Full-scale (subscript f) | Unit | Ratio (ideal, simplified) | Ratio (true scaling) |
|---|---|---|---|---|---|---|
| Geometric scaling factor | $n_l$ | | | - | 1/181 | 1/181 |
| Time scaling factor | $n_t$ | | | - | 1/114 | 1/114 |
| Rotor diameter | $D$ | 1.8 | 326 | m | $n_l$ | $n_l$ |
| Rotor speed at $u_{ts}$ | $\omega_{trans}$ | 501.156 | 4.386 | rpm | $1/n_t$ | $1/n_t$ |
| Rotational inertia | $I$ | 0.036 | $1.232 \times 10^9$ | kg m$^2$ | $n_l^5$ | $n_l^5 \cdot 5.7$ |
| Pitch rate | $\dot{\beta}_{pitch}$ | 85 | 3 | $^\circ\,s^{-1}$ | $1/n_t$ | $1/n_t \cdot 0.25$ |
| Max. aerodynamic torque | $M_{a,max}$ | 5.2831 | $2.7563 \times 10^7$ | Nm | $n_l^3$ (*) | $n_l^3 \cdot 1.14$ |
| Generator torque rate | $\dot{M}_g$ | 28.84 | $1.5 \times 10^6$ | Nm s$^{-1}$ | $n_l^3/n_t$ | $n_l^3/n_t$ |
| Generator torque actuation time (zero to max. torque) | $t_{Mg}$ | 0.1832 | 18.38 | s | $n_t$ | $n_t \cdot 1.14$ |
| Gust duration | $t_{gust}$ | 0.75 | 10.5 | s | $n_t$ | $n_t \cdot 8.14$ |
| Time constant LW mode (at $u_{ts}$) | $T_{LW}$ | 0.2259 | 13.35 | s | $n_t$ | $n_t \cdot 1.93$ |
| Time constant SW mode (at $u_{te}$) | $T_{SW}$ | 0.2812 | 22.21 | s | $n_t$ | $n_t \cdot 1.44$ |

\* further clarification is given in the Appendix A.

### 2.1.2 Torque controller

The tasks of the torque controller are fourfold, i.e., to maintain the minimum rotational speed ($\omega_{min}$) close to cut-in wind speed, to follow the high TSR in the light-wind mode, to keep the rotational speed constant in the transition region ($\omega_{trans}$) and to follow the low TSR in the strong-wind mode. TSR tracking is usually accomplished by setting the generator torque according to the $k\omega^2$-law, as described by Bossanyi (2000), while maintaining a constant rotational speed is a typical task for a PI-controller. Switching of controller types often leads to adverse transient controller reactions. Therefore, the Hybrid-Lambda torque controller is realized with one single PI-controller over the entire partial load range, but the desired behaviour is accomplished by changing the saturations and set points. Figure 2 a shows the rotational speed and generator torque over the wind speed. However, the wind speed is generally not known by the torque controller. Instead, the input is the rotational speed and the output is the generator torque. This schedule (generator torque over rotational speed) is acquired a priori with blade element momentum theory (BEM) simulations and is stored in a LUT. To distinguish between the different LUTs used in this study, we refer to the table mapping generator torque over rotational speed as LUT1. Note, that the pitch angle is varying in the strong-wind mode and the needed generator torque to track the low TSR is subject to model uncertainties. When we plot the generator torque over the rotational speed (compare with Fig. 2) for the Hybrid-Lambda Rotor this function is ambiguous at $\omega_{trans}$ when the controller should switch from the light-wind to the strong-wind mode or vice versa. E.g. multiple generator torque values are assigned to the same rotational speed $\omega_{trans}$. This is where the PI-controller is needed. To describe how the



saturations and set points are derived, we introduce two rotational speeds that we chose from the middle of the light-wind and strong-wind mode, respectively:

$$\omega_{\text{middle,LW}} = \frac{\omega_{\min} + \omega_{\text{trans}}}{2} \tag{11}$$

$$\omega_{\text{middle,SW}} = \frac{\omega_{\text{trans}} + \omega_{\max}}{2} \tag{12}$$

If the rotational speed is between $\omega_{\min}$ and $\omega_{\text{middle,LW}}$, the reference rotational speed is set to $\omega_{\min}$, the lower saturation is set to zero generator torque and the upper saturation is set to the LUT1 (blue line in Fig. 2b). This allows the torque controller to choose generator torques from the blue shaded area in Fig. 2b. But since the set point is $\omega_{\min}$, the generator torque will always follow the upper saturation (i.e. the LUT1 for the high TSR), as desired. Only if the rotational speed gets too low, the PI-controller becomes active and keeps the rotational speed constant at $\omega_{\min}$. Similar methods are applied vice versa if the rotational speed is between $\omega_{\text{middle,SW}}$ and $\omega_{\max}$, although the blue shaded area in Fig. 2 is quite small in this case.

If the rotational speed is between $\omega_{\text{middle,LW}}$ and $\omega_{\text{middle,SW}}$ the reference rotational speed is set to $\omega_{\text{trans}}$, the lower saturation is set to the LUT1 from the light-wind mode (blue line in Fig. 2b) and the upper saturation is set to the LUT1 from the strong-wind mode (red line in Fig. 2b). This allows the torque controller to choose generator torques from the red shaded area in Fig. 2b. If the rotational speed is below $\omega_{\text{trans}}$ the torque controller wants to increase the rotational speed and chooses the generator torque from the lower saturation, consequently tracking the high TSR in the light-wind mode. If the rotational speed is above $\omega_{\text{trans}}$ the torque controller wants to decrease the rotational speed and chooses the generator torque from the upper saturation, consequently tracking the low TSR in the strong-wind mode. Only if the rotational speed is very close to $\omega_{\text{trans}}$ the PI-controller keeps the rotational speed constant, as desired. In other words, the PI-controller is always active. It either succeeds in maintaining $\omega_{\text{trans}}$ or (if it does not succeed due to the saturation limits) it tracks the two different TSRs.





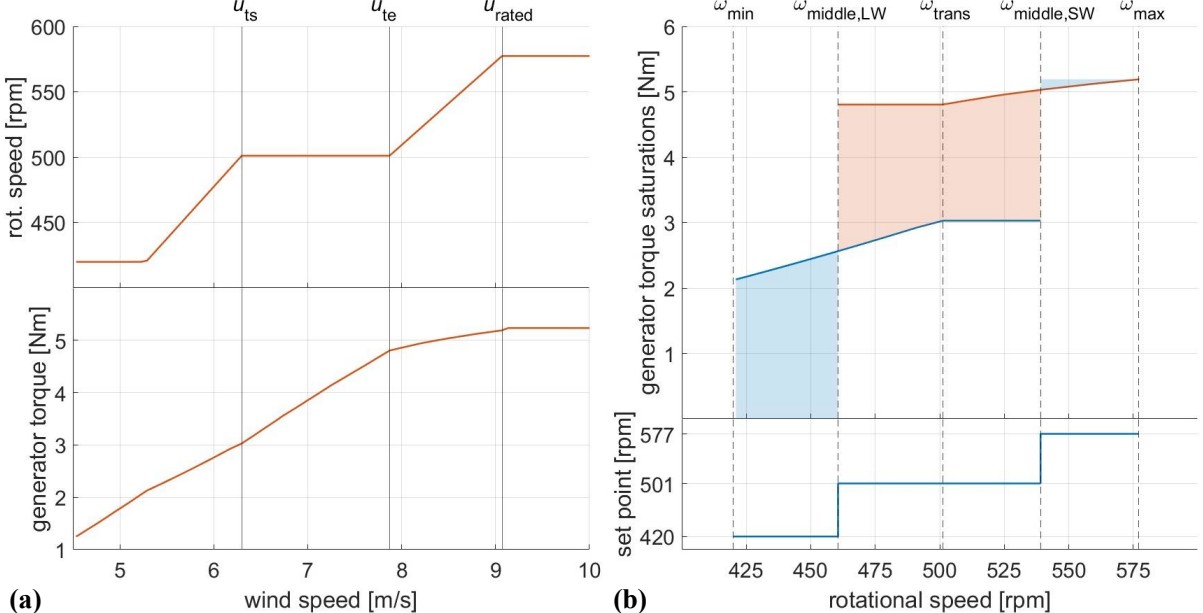

**Figure 2. (a)** Rotational speed and generator torque as a function of wind speed, derived from BEM simulations. **(b)** Saturations and set-points for the torque controller. Shaded areas indicate the permissible generator torque for a given rotational speed.

### 2.1.3 Pitch controller

The pitch controller of a conventional wind turbine has the primary task to maintain a constant rotational speed above rated power. In the Hybrid-Lambda control methodology, it is also used to constrain the maximum loads below rated power. In this paper, we focus on the partial load range and we discuss how the pitch angle can be set between $u_{\mathrm{ts}}$ and $u_{\mathrm{rated}}$ in order to constrain the maximum flapwise RBM. We introduce four versions of the pitch controller - a baseline controller, a load feedback (LFB) controller and a feed-forward (FF) controller. Additionally, we tested a combination of the latter two versions (FFLFB). All four versions are based on a PI-controller, tracking the maximum rotor speed ($\omega_{\mathrm{max}}$) as a reference. But, the minimum pitch angle ($\beta_{\mathrm{min}}$) for limiting the RBM is determined in different ways.

The **baseline controller** (shown in Fig. 3) uses a load-based LUT2 which maps the pitch angles to the respective wind speeds. The derived pitch angle is then set as minimum pitch for the PI-pitch controller. The LUT2 is calculated offline using steady-state measurement data from previous wind tunnel campaigns and data from BEM simulations. Measurement data was available for pitch angles up to $4°$. For larger pitch angles, the dataset was extended with simulations. The baseline controller also includes a wind speed estimator, similar to that of a full-scale turbine. It is based on filtered measurement signals from the rotor speed ($\omega$), the pitch angle ($\beta_{\mathrm{pitch}}$) and the measured low-speed shaft torque ($M_{\mathrm{LSS}}$). The wind speed ($u$) can be derived





from the equations:

$$0 = \frac{1}{2}\rho\pi R^3 u^2 c_{\mathrm{q}}(\beta_{\mathrm{pitch}}, \lambda) - M_{\mathrm{LSS}} - M_{\mathrm{l}}(\omega) - I\dot{\omega} \tag{13}$$

Here, $M_{\mathrm{l}}$ are the friction losses in the drive train which are estimated from a linear fit using measurement data where the drive
train was rotated without the rotor blades, which Berger (2022) describes in his thesis appendix. We found that the inclusion of
the rotor inertia only marginally improved the transient behaviour of the wind speed estimator for this particular wind tunnel
model and we decided to exclude the inertia term for the sake of simplicity. A comprehensive description of more sophisticated
wind speed estimators is given among others by Soltani et al. (2013) and by Knudsen and Bak (2013). The torque coefficient as
a function of pitch angle and TSR was also derived from steady-state measurement data from previous wind tunnel campaigns.
It was extended with data from BEM simulations for combinations of pitch angles and TSRs where no measurement data was
available. The objectives of this pitch controller is to set up a relatively simple baseline controller using only standard sensors
that are also available on a conventional full-scale wind turbine. It further serves as a reference for the advanced controllers,
which are described in the next paragraphs.

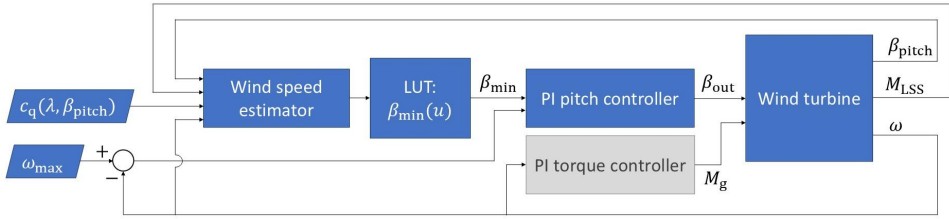

**Figure 3.** Baseline pitch controller. Torque controller indicated in simplified form. Rounded rectangles: Input/output of (measured) variables.
Rectangles: Processes. Parallelograms: Data input.

The **load feedback controller (LFB)** (shown in Fig. 4) uses measurements from the strain gauges at the blade root to set
the individual pitch angle. The measurement signal from the flapwise RBM is low-pass filtered, the maximum allowable load
is subtracted and the residual is converted with a positive controller parameter to the pitch demand. This results in a pitch
increment ($\Delta\beta_{\mathrm{pitch}}$) which can either be positive or negative, but it is saturated with the maximum allowable pitch rate. It is
then added to the previous pitch angle ($\beta_{\mathrm{pitch,i\text{-}1}}$) and if the result is larger than the minimum pitch from hardware constraints,
it is set as the minimum pitch angle for the PI-pitch controller. This implementation represents an integral behaviour which
has the disadvantage of slower system response times. The measurement signal from the flapwise RBM needs to be low-pass
filtered because the noise in the measurement data would lead to an undesirable fluctuating controller response. We had to use
a cut-off frequency of 1 Hz (which corresponds to about eight rotor revolutions at $\omega_{\mathrm{trans}}$) to yield a smooth controller reaction.
The low-pass filter introduces further delay and additionally slows down the system response time. The described controller
logic is performed for each blade individually. But, due to the slow response time, this controller does not represent a classical
individual pitch controller and 1P blade load variations can not be alleviated. It behaves like a collective pitch controller with a
rotor balancing component, which reacts on long time scales. The advantages of the LFB controller are that it is not dependent





on model accuracy, there is no need for a wind speed estimator and no a priori calculation of the pitch angle (LUT2) is required. The objective of this controller is to accurately constrain the mean value of the flapwise RBM.

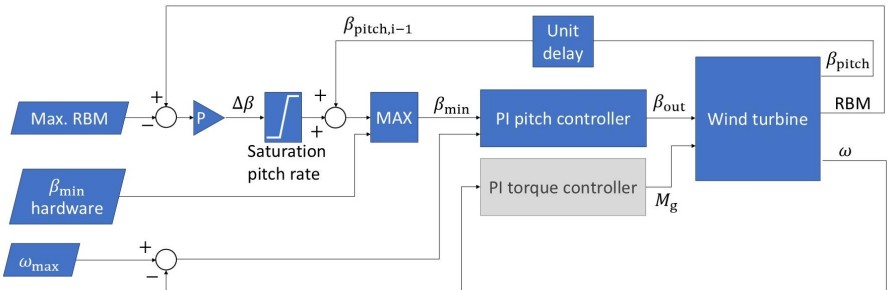

**Figure 4.** Load feedback (LFB) pitch controller. Torque controller indicated in simplified form.

The **feed-forward controller (FF)** (shown in Fig. 5) uses an upstream wind speed measurement to compensate for the wind
disturbance. In the wind tunnel, single point hot-wire measurements at 1.44 diameters in front of the rotor area are used to emulate a simple single beam staring mode nacelle lidar. The wind speed measurements are processed and propagated to the rotor area, as further explained in the next paragraph. The signal is then fed to the LUT2 to determine the required pitch angle for the respective wind speed which is set as minimum pitch angle for the PI-pitch controller. In general, the objective of this controller is to act in advance rather than reacting and specifically to reduce load overshoots in gust events.

We additionally implemented a **combined feed-forward load feedback controller (FFLFB)**. Here, the logic of both the FF and the LFB controllers run in parallel and always the larger output pitch angle is used. In other words, $\beta_{\mathrm{min}}$ from the FF controller (Fig. 5) replaces $\beta_{\mathrm{min,hardware}}$ in the logic of the LFB controller (Fig. 4). The FFLFB controller was only evaluated for the gust test case and the waked inflow.

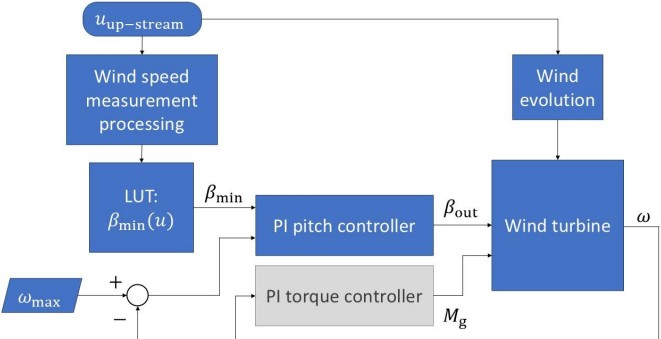

**Figure 5.** Feed-forward (FF) pitch controller. Torque controller indicated in simplified form.

To process the upstream hot-wire measurements for the FF controller, the signal needs to be delayed to account for the
propagation time from the measurement location to the rotor plane (see Fig. 6), under the assumption of the frozen turbulence





hypothesis. In case of a gust event, ideally, the pitch angle would start to increase at the exact time when the gust would hit the rotor. To account for the delay in the pitch actuation system, the controller would need to increase the reference signal slightly in advance. If we even want to account for some safety margin considering model uncertainties and the stochastic nature of turbulent gusts we would rather like to start pitching a bit too early than too late, e.g. starting to pitch just before the gust

arrives. The simplest solution would be to decrease the propagation time. In this case, the pitch would increase just before the gust arrives but it would also start to decrease just before the gust has ended, which would lead to a load overshoot at the end of the gust event. Thus, we need to process the measurement signal in a way that the gust duration is enlarged. In other words, we would like to detect up-ramps early (increasing the pitch angle just before the gust arrives) and delay down-ramps (decreasing the pitch angle just after the gust has finished).


Therefore, the hot-wire measurement signal is processed as illustrated in the flow chart in Fig. 6. An exemplary wind speed time series is plotted in Fig. 7. First, the signal is low-pass filtered (3 Hz cut-off frequency) to reduce noise (red line in Fig. 7a). Second, the signal is delayed by a fixed, short delay time (yellow line in Fig. 7a) and subtracted by the non-delayed signal, resulting in $\Delta u(t)$. This gives us an indication about the expected change in the signal. Third, $\Delta u(t)$ is converted with a

controller parameter and added to a constant delay time that accounts for the time lag of the low-pass filter, the propagation time for stationary inflow and the delay in the pitch actuation system. Fourth, the original (filtered) signal is then delayed with this variable delay time and stored in a buffer from which the controller reads a value for each controller update iteration (yellow line in Fig. 7b). To visualize the effect of the processing, the resulting signal is plotted without propagation in Fig. 7c. Similar results could be achieved with a moving average or a Gaussian weighted moving average filter. Alternative methods

and investigations on the propagation time for a model predictive controller for wind turbines in wind tunnel experiments are provided by Sinner et al. (2023).

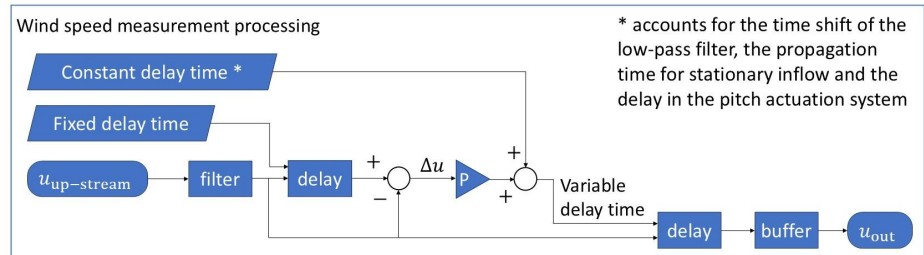

**Figure 6.** Wind speed measurement processing for the FF pitch controller.



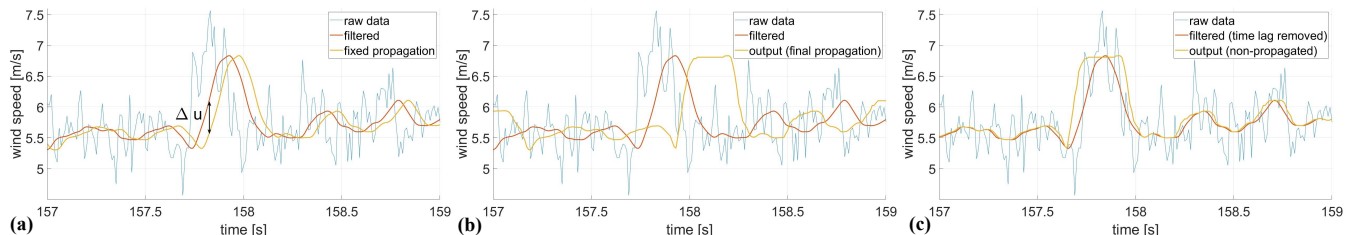

**Figure 7.** Exemplary wind speed measurement processing for the FF pitch controller for a gust event. **(a)** Raw and filtered data, **(b)** final propagation and **(c)** final signal with removed propagation for visualisation purpose.

## 2.2 Measurement set-up and post-processing

The experimental controller validations were carried out in August 2024 in the turbulent wind tunnel in Oldenburg with a cross section of $3 \times 3$ m, which is described by Kröger et al. (2018). The experimental set-up is sketched in Fig. 8. The open jet

configuration with the active grid (Neuhaus et al., 2021) is used to investigate the controllers under turbulent reproducible inflow test cases. The control algorithms are implemented for the Model Wind Turbine Oldenburg (MoWiTO 1.8) with a diameter of 1.8 m, further described by Berger et al. (2018), equipped with blades designed by Ribnitzky et al. (2025) according to the Hybrid-Lambda rotor design methodology. Measurement data used for the present study comprise rotational speed (azimuthal encoder), low-speed-shaft torque (torque meter), flapwise blade root bending moments (strain gauges) and upstream wind speed

measurements (one-dimensional hot-wire), each sampled at 10 kHz on a National Instruments Compact RIO data acquisition system. The control algorithms are implemented on the aforementioned device with a LabVIEW real-time application. The controller update frequency is set to 100 Hz due to limitations in the computational performance of the real-time system.

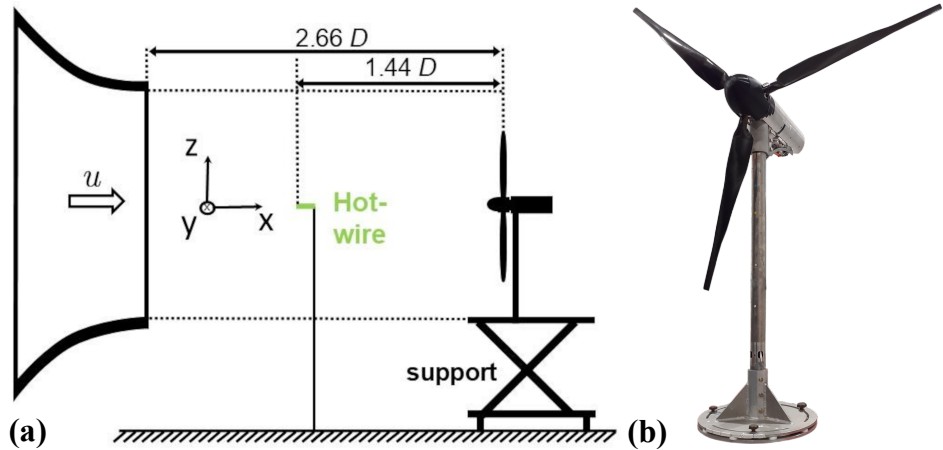

**Figure 8. (a)** Experimental set-up, side-view. **(b)** MoWiTO 1.8 with the Hybrid-Lambda blades, diameter of 1.8 m, hub height of 1.5 m.





The measurement data is post-processed as follows. The torque measurements are corrected for the friction in the drive train (bearings and slip ring) in order to deduce the aerodynamic rotor torque from the measured low-speed-shaft torque. To filter high frequency vibrations and noise out of the measurement signals, we use zero-phase digital filters with a Butterworth low-pass filter in the post-processing. The corrected torque and power signals are then low-pass filtered with a half-power-frequency of 10 Hz, the flapwise RBM measurements with 28 Hz which is 3.4-times higher than the rotational frequency (1P) of 500 rpm. This ensures that all relevant rotor dynamics are resolved. In general, we show the mean of all three blade root bending moments. The signal of the rotational speed is smoothed with a moving average window of 0.5 s length. The TSR is calculated from the filtered rotational speed and the wind speed estimate, which is also smoothed with a moving average window of the same length. The statistics of the aforementioned post-processed data is illustrated in box plots in Fig. 9 b and c. Here, the middle markers show the median, boxes represent the 25th and 75th percentiles and the whiskers extend to the minimum and maximum of the considered data. Statistics shown in the box plots are evaluated over the stationary last 10 s of each wind step. For the time series figures, the RBM signals are filtered with a half-power-frequency of 5 Hz, to increase the clarity. For the gust events, we show ensemble averages of the 50 gust repetitions with the 95% confidence interval indicated with the shaded area. Here, we use half-power frequencies of 28 Hz for the RBM signals, 10 Hz for the corrected torque and 5 Hz for the hot-wire wind speed measurements.

### 2.3 Test cases

The controllers are experimentally investigated with four different test cases. First, wind steps are used to validate the different controller implementations. The flow is uniform, the active grid is installed with all flaps in static open configuration which leads to a background turbulence intensity of 2%. The wind speed is increased every 20 s by increasing the rotational speed of the wind tunnel fan.

The second test case represents coherent gust events, generated with the active grid, mimicking a scaled extreme operating gust with the so-called Mexican hat shape, according to IEC 61400-1 (2019). This inflow test case was characterized by Neuhaus et al. (2021) and it was further used in our previous publication (Ribnitzky et al., 2025) to compare the effect of different blade designs on the flapwise RBM. But in the latter case, no wind turbine controller was used and the rotational speed was controlled to be constant since the emphasis on the previous study was on the different blade designs. In contrast, in this study, different controller versions are compared for the same gust events.

The third test case is a turbulent inflow, mimicking the design load case 1.6 (DNV-GL ST-0437, November 2016), normal power production with the normal turbulence model, which was also investigated with aero-servo-elastic simulations on the full-scale Hybrid-Lambda Rotor (Ribnitzky et al., 2024). The background turbulence is produced with the active grid, while the large wind speed fluctuations on the slow time scale are realised by dynamically changing the rotational speed of the wind tunnel fan. The mean wind speed (derived from the wind speed estimator) ranges from 5 to 11 m s$^{-1}$. The turbulence intensity (derived from the hot-wire measurements) is 15%. Due to the inertia of the airflow and the wind tunnel drive sys-



tem the wind speed can not be changed infinitely fast without compromising the wind speed amplitude. Therefore, this test case is not properly scaled in time. As previously described, the time scaling factor for the Hybrid-Lambda model turbine is $n_\mathrm{t} = t_\mathrm{m}/t_\mathrm{f} = 1/114$ to correctly represent the 15 MW full-scale equivalent with a diameter of $D_\mathrm{f} = 326\,\mathrm{m}$. Here, the subscripts

m and f refer to the model turbine and full-scale turbine, respectively. This means, the 600 s load case with all the wind speed variations would need to be scaled to 5.3 s. Unfortunately, this is not possible if wind speed amplitudes of up to $6\,\mathrm{m\,s^{-1}}$ are desired in order to cover the whole operating range of the model turbine. This load case is therefore not scaled in time but focuses on the ability of the controllers to switch the operating modes in turbulent inflow conditions. The fast dynamics are better described by the scaled gust events mentioned before.


The fourth test case represents waked inflow conditions. A velocity deficit profile is produced with the active grid, mimicking a wake of an imaginary upstream turbine that meanders in horizontal and vertical directions on the basis of an Ornstein-Uhlenbeck process, following a method described by Onnen et al. (2024). The produced wake deficit shows a nearly axi-symmetric Gaussian profile and the spectral characteristics (e.g. the wake meandering frequency) is true to scale compared

with a full-scale turbine. This means, the cut-off frequency in the dynamic wake meandering model $f_\mathrm{c} = u_\infty/(2D) = 1.67\,Hz$ is representative for the wind tunnel model. The mean rotor effective wind speed in full-wake operation is set to approximately 6 m/s, which means the turbine operates mostly in light-wind mode. For large wake centre offsets, the rotor effective wind speed increases and the model turbine operates in the transition region, impinged by a partial wake scenario.

## 2.4    Calculation of pitch actuation and blade fatigue loads

Several advanced control algorithms can provide benefits in terms of power output or reduction of extreme loads, but they often come with the drawback of increased actuation activity. This can lead to increased wear and the need for higher maintenance intervals or even component replacements can outweigh the benefits. Therefore, we calculate the pitch actuator duty cycle (ADC), as described by Bottasso et al. (2013), according to Eq. 14.

$$\beta_\mathrm{ADC} = \frac{1}{T}\int_0^T \frac{|\dot{\beta}_\mathrm{pitch}|}{\dot{\beta}_\mathrm{pitch,max}}\,\mathrm{d}t \tag{14}$$

Here, $\dot{\beta}_\mathrm{pitch,max}$ is the maximum allowable pitch rate. We further analyse the fatigue loads in terms of damage equivalent loads (DEL) for the blade root bending moments. The 28 Hz low pass filtered RBMs are processed, using a rainflow counting algorithm by DOWNING and SOCIE (1982) with a Wöhler curve exponent of 10 for composite materials. Note, that the DELs are simply calculated for the time series of the individual test cases to express the impact of the control algorithms on the short-term fatigue loading. These numbers can not be related to lifetime damage equivalent loads since no wind speed probability

distribution is used and not all design load cases that are required by means of the certification standards are considered.



## 3  Results

In this section, we analyse the performance of the different controller versions with the four test cases. Besides extreme loads, power output and the switching between the operating modes, we elaborate on pitch actuation and fatigue loads.

### 3.1  Wind steps

We first validate the controller implementations with wind steps. The time series of the turbine responses are shown in Fig. 9 a. The expected set points are shown with the black lines and the expected operating modes are indicated with the background colours. Those are derived from the steady-state operating points, i.e. the data shown in Fig. 1, using the rotor effective wind speed calculated with the estimator. The baseline controller works well in controlling the rotational speed and the TSR-change, i.e. following a TSR of 7.5 in the light-wind mode, maintaining a constant rotational speed of 500 rpm in the transition region

and following a TSR of 6 in the strong-wind mode. In general, the loads are higher than expected in the partial load range ($60\,\mathrm{s} < t < 140\,\mathrm{s}$). For 6 m/s wind speed ($140\,\mathrm{s} < t < 160\,\mathrm{s}$), we still expect the rotor to be in light-wind mode, meaning the pitch angle is at minimum pitch and the loads are expected to be below the constraint of 7.3 Nm. But, the loads already exceed the limit which can not be recognized by the baseline controller. For higher wind speeds ($160\,\mathrm{s} < t < 260\,\mathrm{s}$) in the transition region and in the strong-wind mode, the baseline controller starts to increase the pitch angle and succeeds to keep the RBM at a

constant level. However, the mean value is higher than the set point (8 Nm instead of 7.3 Nm) due to model uncertainties in the LUT2. Possibly, the zero pitch angle was set to a slightly lower pitch angle when evaluating the controllers in this measurement campaign, compared to the experiments when the data was recorded to set up the LUT2 which were performed in 2023. Such an offset in the pitch angle could explain the increased loads.

Here, the advantages of the LFB controller become clear. With the load feedback, the controller corrects for the offset in the model and is able to correctly constrain the flapwise RBM. However, this comes with the cost of reduced power output. This is especially noticeable in the strong-wind mode ($220\,\mathrm{s} < t < 260\,\mathrm{s}$). Moreover, the TSR in the strong-wind mode is lower than the set point (5.7 instead of 6) for the LFB controller ($220\,\mathrm{s} < t < 260\,\mathrm{s}$). A higher pitch angle is used, which successfully constrains the loads, but this also reduces the aerodynamic torque. Consequently, the reference generator torque is too high and

the rotational speed as well as the TSR are lower than the set point. It can be seen that the LFB controller uses lower generator torque values compared to the baseline controller, since the reduced rotational speed moves the set point in the LUT1 to lower values. Ideally, the LUT1 itself would need to be adapted to account for the higher than expected pitch angles. This could be realized by calculating the generator torque with the $k\omega^2$-law, using the torque coefficient surface and the actual pitch angle on the fly, rather than using the a priori calculated LUT1 with expected pitch angles.


The FF controller behaves similarly to the baseline controller. But, it uses single-point hot-wire measurements instead of the wind speed estimator and due to the aforementioned model uncertainties the commanded pitch angle differs from the baseline controller. Since there is no feedback from the loads, the FF controller fails to keep the RBM constant in the transition and



strong-wind mode. The advantages of the FF controller will become more clear in Sect. 3.2 which addresses the gust events.

The statistical analysis of the wind steps is shown in Fig. 9 b and c. The median is shown with the middle marker, the 25th and

75th percentiles are visualized with the boxes and the whiskers extend to the extrema of the data.



**Figure 9.** Turbine response with the baseline, LFB and FF controllers to the test case with wind steps: Time series **(a)** and statistical analysis of RBM **(b)** and power **(c)**. Background colours indicate the operating mode: green, light-wind mode; yellow, transition; red, strong-wind mode. Middle markers, median; boxes, 25th and 75th percentiles; whiskers, minimum and maximum of the considered data.



## 3.2 Gusts

We use gust events to investigate the transient behaviour of the controllers. The ensemble averages of the 50 gust repetitions are shown in Fig. 10. The wind speed measurements are taken 1.44 D (2.7 m) upstream of the rotor and the data is not propagated

in the plot. Since only single point measurements are available, an offset to the rotor equivalent wind speed is expected and the load level does not perfectly match with the data from the wind steps test case. However, this does not affect the presented analysis of the transient behaviour. In Fig. 10 the 95% confidence intervals of the 50 gust repetitions are indicated with the shaded areas. The narrow width of the confidence intervals highlights the good repeatability of the individual gust events.

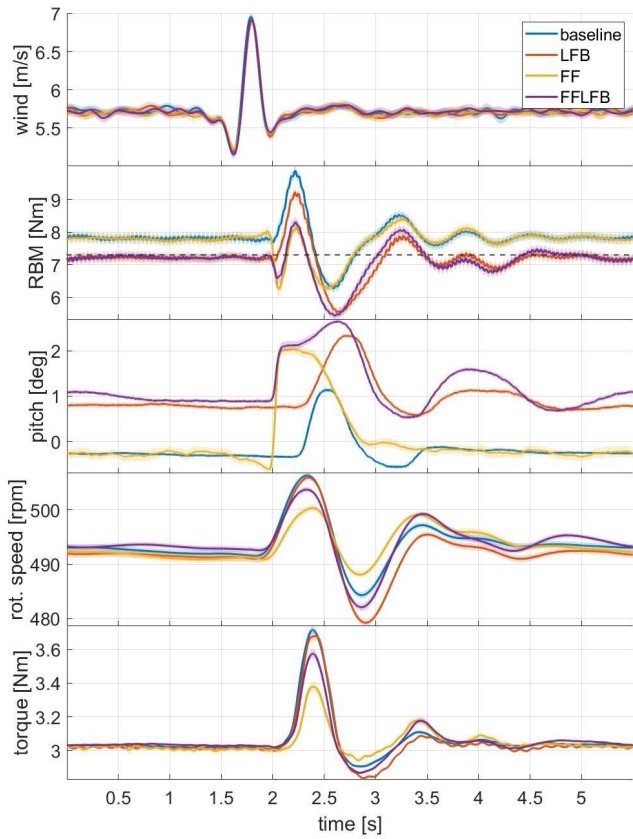

**Figure 10.** Turbine response to gust events. Solid lines show the ensemble average over 50 gust repetitions. The shaded areas indicate the 95% confidence intervals. Wind speed measurements are performed 2.7 m upstream of the rotor and are not propagated in this plot. The dashed line represents the load constraint of 7.3 Nm.

First of all, we notice the different steady-state values in the flapwise RBM. The controllers with the load feedback (LFB

and FFLFB) can limit the loads perfectly to the constraint of 7.3 Nm in steady-state. In contrast, the loads are higher than the set point for the baseline and the FF controller, as observed for the wind steps as well. During the gust event, the baseline controller shows the highest overshoot in the flapwise RBM. The maximum load is 25% higher than the steady-state RBM,





because the baseline controller increases the pitch angle too late. The LFB controller shows the slowest reaction, due to the inherent integral behaviour of the controller architecture and due to the time delay associated with the low-pass filtering of the load measurements. In fact, the applied filter introduces a delay of $0.2\,\mathrm{s}$ for the given gust events. In contrast, the FF controller reduces the peak load almost to the same level as the steady-state loads. The feed-forward controllers (FF and FFLFB) react in advance and increase the pitch angle just before the gust arrives. This reduces the flapwise RBM in advance and once the gust arrives, the load overshoot is significantly reduced. More precisely for the FF controller, the peak load is 4% higher than the steady state value, but still 11% higher than the load constraint. The feed-forward controllers also show a longer period of high pitch angle, compared to the baseline controller. This is the intended behaviour resulting from the processing of the feed-forward signal, as described in Sect. 2.1.3. Further, the FF controller effectively reduces the variations in the rotational speed and the generator torque which is beneficial for the drive train loads and leads to a more even power feed-in. As intended, the FFLFB controller combines the advantages of the FF and the LFB controllers. It can constrain the steady-state mean load to the desired value due to the loads feedback and additionally, it can react in advance due to the preview information. For the FF controller, the lack of load feedback is further visible, since the load overshoot after the gust event (at $3.25\,\mathrm{s}$) is 4% higher than during the gust event (at $2.25\,\mathrm{s}$).

Only the FF controller reached the maximum pitch rate of $85\,°/s$, which happened in 14 of the 50 investigated gust events. For the ensemble average, the highest pitch rate observed was $46\,°/s$ for the FF controller. The other controllers operated far from the hardware limit. In comparison, the highest pitch rate observed for the baseline controller was $18\,°/s$. The maximum torque rate observed was $8.4\,\mathrm{Nm\,s^{-1}}$ for the baseline controller and it was similar for all four controller versions. Consequntly, less than one third of the maximum generator torque rate was used.

### 3.3 Turbulent wind fields

With this test case, we want to evaluate the controllers on their ability to switch between the operating modes in varying turbulent inflow conditions. The results are plotted in Fig. 11. The desired operating mode is derived on the basis of the wind speed estimator which is indicated with the background colours. As for the previous test cases, the baseline and FF controllers set too low pitch angles and the flapwise RBM exceeds the constraint, due to the model mismatch. In the transition and strong-wind modes, the average flapwise RBM is 6% higher than the constraint for the baseline controller and 9% higher for the FF controller. In contrast, the LFB controller can limit the loads effectively, due to the load feedback. However, the increased pitch angle also reduces the power output. On average over the entire duration of the test case, the power output of the LFB controller is 5% lower than for the baseline controller. However, the power reduction is expected due to the controller design, which favours load limitation over power maximisation.

All controller versions succeed in tracking the TSR of 7.5 in the light-wind mode. Although, the LFB controller seems to reach the desired TSR last when transitioning from strong-wind to light-wind mode (e.g. at $215\,\mathrm{s}$, $260\,\mathrm{s}$ and $380\,\mathrm{s}$). As in the wind steps test case, the baseline controller works best in regulating the TSR at 6 in the strong-wind mode. In contrast, the



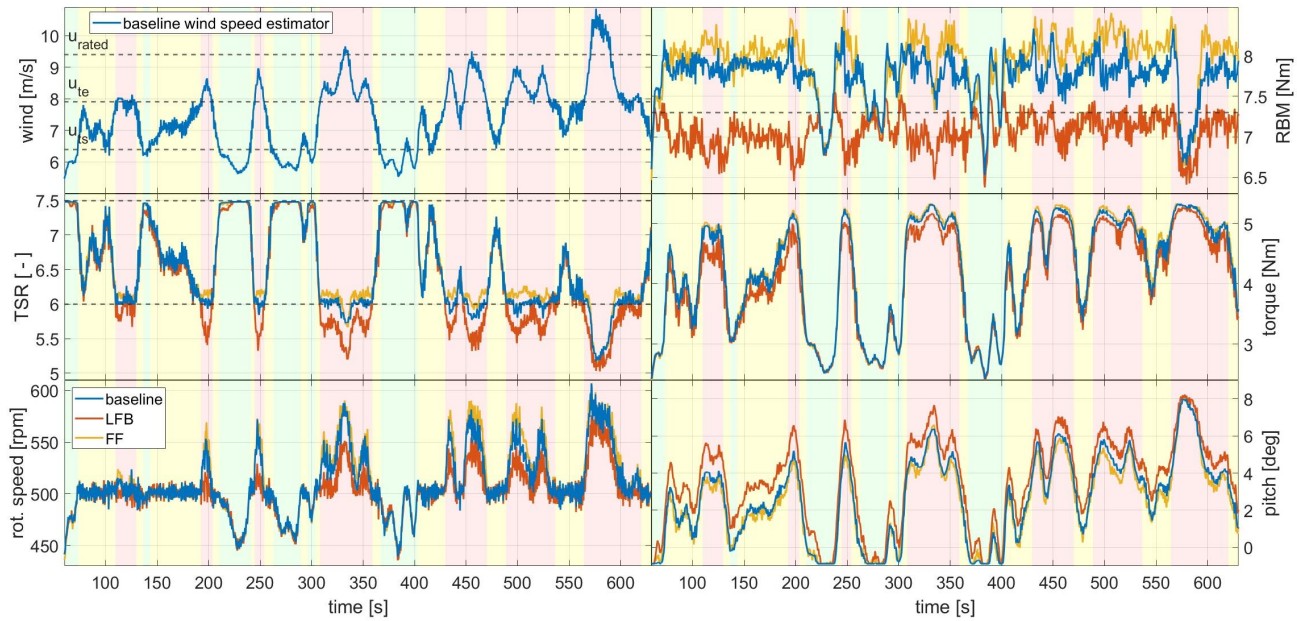

**Figure 11.** Turbine response to turbulent inflow. Background colours indicate the desired operating modes: Green, light-wind; red, strong-wind; yellow, transitioning between the operating modes.

TSR is lower than the set point for the LFB controller in strong-wind mode. Higher pitch angles are set which reduces not only the flapwise RBM but also the aerodynamic torque and therefore the rotational speed. In general, the torque controller works well in maintaining $500\,\text{rpm}$ in the transition region. Between $570\,\text{s}$ and $590\,\text{s}$, the wind speed exceeds rated. Here, the pitch
controller maintains the maximum rotational speed and the TSR drops below 6 for all controllers, as expected.

For this test case, the FF controller used the highest pitch rate of $49\,°/s$ which is only 60% of the hardware constraint. The maximum pitch rate for the other three controllers are below $25\,°/s$. The maximum torque rate observed was $8.1\,\text{Nm s}^{-1}$ for the RBM controller.

**3.4 Waked inflow**

This test case addresses non-uniform inflow and we evaluate the controller versions under a meandering wake inflow scenario. The wake produced with the active grid meanders in horizontal and vertical directions as a random walk. The radial distance of the wake centre to the rotor centre is plotted in Fig. 12 as an absolute value. We compare this signal to the hot-wire measurements on the centre line and the wind speed estimator. Once the signals are propagated they correlate well in time.
However, the single point hot-wire measurements fail to provide a representative rotor effective wind speed. Whenever the wake centre distance is below 0.5 D, the hot-wire measurements are lower than the estimated wind speed and all parts of the rotor that are in the free stream are not considered by this measurement technique. Only for very large wake centre offsets (e.g.





$440\,\text{s} < t < 450\,\text{s}$), the hot-wire measurements converge closer to the signal of the wind speed estimator. The latter works much better in providing a rotor effective wind speed since the rotor is used as a sensor to capture all effects over the rotor area.

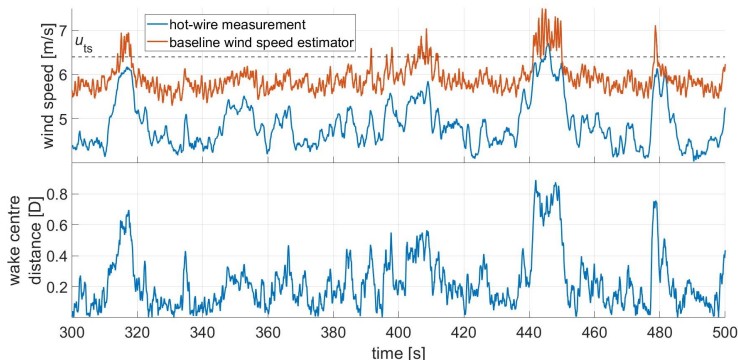

**Figure 12.** Top: Wind speed derived from hot-wire measurements on the centre line (propagated) and derived from the wind speed estimator. Bottom: Radial distance of the wake centre to the rotor centre.


The average of the rotor effective wind speed over the duration of the test case is set to $6\,\text{m/s}$ which means the turbine is in light-wind mode. The time intervals where the wake centre offset is large enough that the rotor operates in the transition region are of highest interest, as shown in Fig. 13. All controller versions succeed to track the light-wind mode TSR. Further, the reduction in TSR in the transition region works equally well for all versions. Differences are visible in the load limitation.

The FF controller is blind to any non-uniformity in the flow and for large wake centre offsets, the exceedances of the flapwise RBM are most pronounced (e.g. $t = 315\,\text{s}$, $t = 410\,\text{s}$, $t = 450\,\text{s}$). In contrast, the baseline controller uses the estimated rotor averaged wind speed and chooses higher pitch angles which leads to reduced load overshoots. The LFB and FFLFB controllers again take advantage of the loads feedback and perform best in limiting the flapwise RBM. In summary, the maximum load exceeds the constraint of $7.3\,\text{Nm}$ by 17% for the FF controller, by 12% for the baseline, by 6% for the FFLFB and by 5% for

the LFB controller. Note, that the load time series presented in Fig. 13 are low-pass filtered with a half-power-frequency of $5\,\text{Hz}$ to increase clarity and $1\text{P}$ variations are filtered out.

Although the FF controller uses the lowest pitch angles and shows the highest loads, the power output is not increased compared to the other controllers. In contrast, the power output of the FF controller shows more fluctuations in the light-wind

mode because the smaller variations in the rotor effective wind speed are not captured by the single point measurements. Consequently, the wind speed is assumed to be below $u_{\text{ts}}$ and the pitch actuation is not triggered (as it is for the other controller versions) which leads to higher power fluctuations (e.g. $320\,\text{s} < t < 405\,\text{s}$).

For this test case, the FF controller used the highest pitch rate of $53\,^\circ/s$, the FFLFB controller used $42\,^\circ/s$ and the baseline

and LFB controller used $19\,^\circ/s$. The maximum torque rate observed was $7.2\,\text{Nm\,s}^{-1}$ for the baseline controller.





**Figure 13.** Turbine response to waked inflow. **(a)** $200\,s$ and **(b)** $60\,s$ excerpt of the full test case. Background colours indicate the desired operating modes: green, light-wind; yellow, transitioning between the operating modes.





## 3.5 Pitch actuation and blade fatigue loads

In the previous sections, we mainly addressed extreme loads and power output for the individual test cases. Finally, we compare all controller versions among all test cases in terms of pitch actuation and blade fatigue loads for the flapwise bending moments, as shown in Fig. 14. For the pitch actuation, the gust events are the most severe load case. To compute the DEL
and ADC all 50 repetitions of the gust event were used, which is not a realistic load case, but it can be used as a metric for comparison within this study. Compared to the baseline and LFB controller, the FF controller leads to around three times higher pitch actuation for the gust test case. As already seen in Fig. 10, the pitch manoeuvre for the FF controller starts with a much higher slope which drives up the pitch actuation value. Further, for the turbulent wind field test case, the FF controller shows the highest actuation among all controller versions, although the value is only 70% of the pitch actuation from
the gust events. The baseline and LFB controller show similar pitch actuations for the gusts and turbulent inflow test cases with a small increase for the turbulent inflow. This can be related to the small scale fluctuations in the inflow which are not present in-between the gust events. Among all performed test cases, the waked inflow leads to the lowest pitch actuations for all controller versions because the turbine is in light-wind mode most of the time with none to little pitch actuation. Here, the FF controller shows rather low actuations, because the pitch is often not triggered. The single-point measurements on the cen-
tre line do not fully capture the variations in the rotor effective wind speed and this also leads to lower values of pitch actuation.

For the flapwise blade fatigue loads, the waked inflow is the most severe test case. This can be related to the non-uniformity of the flow. The blades sample through spacially inhomogeneous wind fields and the individual blade loads are excited with higher harmonics of the blade passing frequency which is not present for the other two test cases with uniform inflow. For the
gust test case, the strategy of the FF controller seems to pay off not only in terms of reduced extreme loads. Also the DELs are the lowest among the four controller versions for this test case. In general, the turbulent inflow test case leads to similar DELs, compared to the gust events. For the turbulent inflow, the LFB controller works best in reducing the extreme loads, but it leads to higher blade fatigue loads (7% increase compared to the baseline controller).





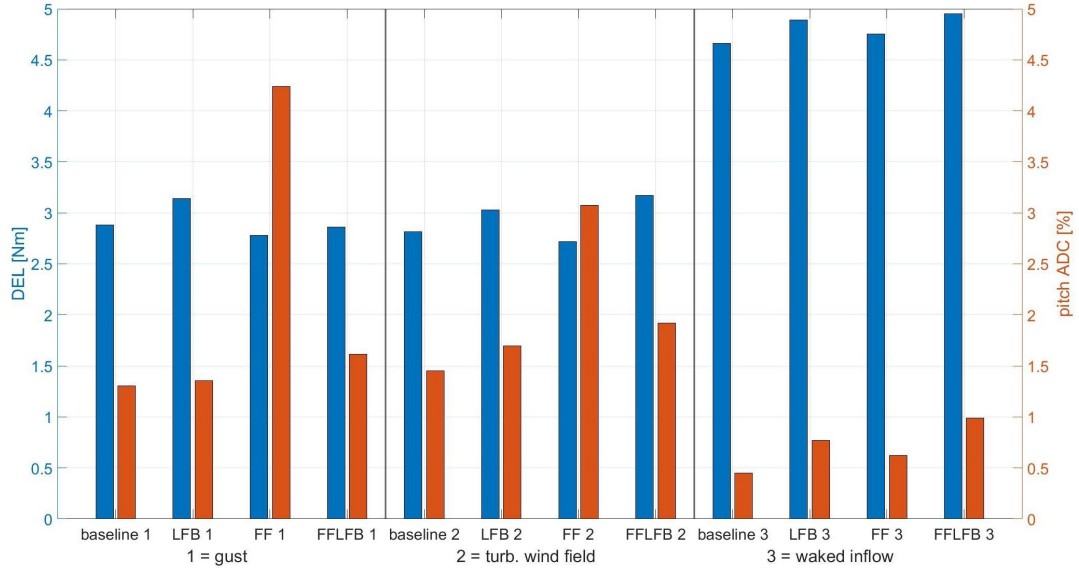

**Figure 14.** DEL and pitch actuator duty cycle for the three inflow cases: Gust, turbulent wind field and waked inflow.

## 4 Discussion

The novelty of the Hybrid-Lambda control strategy is the objective of tracking two different TSRs below rated wind speed and the realization of the transition between the operating modes with an overarching load constraint. Contrary to conventional wind turbine controllers, the pitch and torque controllers are active simultaneously over a wide range of wind speeds below rated. The wind tunnel validation clearly showed that it is possible to realize the Hybrid-Lambda control strategy. The latter was so far only designed with steady-state analysis and tested in transient aero-servo-elastic simulations. With the research presented here, we evaluated the controller methods on a fully actuated scaled wind turbine.

The results confirm the original research questions outlined in the introduction. First, the transition between the operating modes (i.e. keeping the rotor speed constant while the pitch controller is active for the load constraint) can be accomplished by using the torque PI-controller. Despite the large rotor inertia, variations of only +/-2% in rotor speed are present for turbulent inflow when the turbine operates in the transition region. The second objective of tracking a desired TSR and simultaneously achieving a load constraint introduced a challenge to the controller design. In the present study, we tested two controller architectures that can accomplish one of the objectives, but struggle to fulfil both simultaneously. The baseline controller succeeded best in tracking the strong-wind TSR, but due to a model mismatch, led to too high loads. The LFB controller was able to limit the loads to the desired constraint. But, by choosing higher pitch angles, the operating points of the aerodynamic model deviated from the expected ones resulting in a poor tracking of the strong-wind TSR by the torque controller. To overcome this and to achieve both objectives independently of each other, the $k\omega^2$-law in the torque controller would need to be adapted





with a $k$-value that is a function of the actual pitch angle and which needs to be calculated on the fly. The third objective was to limit extreme flapwise RBM loads. If the mean value should be set correctly, a load feedback is necessary and the LFB controller performed best. However, due to its integral behaviour, this controller architecture can not handle transient events on a small time scale like gusts. In this case, the FF controller performed best in limiting the extreme loads. Further, controller designs without additional sensor equipment, like the baseline controller, can keep the loads constant in the transition and strong-wind mode. However, one needs to take model uncertainties into account, which leads us to the fourth objective. Model uncertainties can lead to an offset in the load response as it is present for the baseline controller. The LFB controller can overcome this challenge thanks to the measurement data from the flapwise RBMs. Alternatively, a model calibration or model update procedure would be needed to improve the baseline controller, as among others described by Mulders et al. (2023). Model uncertainties will also be present for a commercial full-scale turbine. The inflow is not controllable nor repeatable in the field. However, the aerodynamic model might be more reliable due to higher chord based Reynolds numbers. Further, the manufacturing and assembling accuracy will be higher for a full-scale turbine because the ratio of tolerances to product size is simply more favourable. With the fifth research question, we addressed further impacts on the wind turbine as an engineering system, such as the pitch actuator duty cycle and fatigue loads. Besides limiting the extreme loads, the FF controller could also reduce the fatigue loads on the blade root bending moments, but comes at the cost of an increased pitch actuation.

Testing wind turbine applications in the wind tunnel always comes with the challenge of appropriate scaling (Canet et al., 2021). While the aerodynamic scaling of the wind turbine model was successfully described by Ribnitzky et al. (2025), the temporal scaling of the inflow conditions remains difficult to satisfy. One limitation of the methodology presented here is that the turbulent inflow test case is not true to scale in time, compared to a full-scale equivalent. The wind speed variations can not be accomplished arbitrarily fast since the flow itself as well as the wind tunnel actuators are subject to inertias. However, we can investigate the transient controller reactions with the test case of gust events. The fastest gust that can be produced with the active grid and that still shows a reasonable wind speed amplitude has a duration of 0.75 s (Neuhaus et al., 2021). Due to the large time scaling factor of the model turbine ($n_{\mathrm{t}} = 1/114$), this gust is still about eight times slower than the full-scale equivalent which should have a duration of 10.5 s according to the IEC 61400-1 (2019). However, for the model turbine, the rotational inertia and the pitch and torque actuation times are larger compared to an ideally scaled version of the full-scale turbine. Consequently, the transient controller reactions can be assessed reasonably well with this compromise.

## 5 Conclusions

In this paper, we set up a control methodology for the Hybrid-Lambda Rotor concept and validated it on a scaled wind turbine model in wind tunnel experiments under various turbulent inflow conditions. We showed how two different TSRs can be tracked below rated wind speed by arranging the rotor speed set points and generator torque saturations. With this method, we can further realize a transition region with constant rotational speed to switch between the two operational TSRs. Four versions





of the pitch controller, which aim to satisfy the constraint on the flapwise RBM, were investigated. First, a baseline controller which uses only standard sensors that are also available on a conventional full-scale wind turbine. It performed well in tracking the different TSRs and managed to constrain the loads to a constant value which was however 10% higher than the desired level (in the test case with wind steps) due to a mismatch in the zero pitch position. Second, a load feedback controller that uses measurements from the strain gauges at the blade root. This version was able to correctly set the mean value of the flapwise

RBM, but reducing the loads always comes with the drawback of reduced power feed-in. Further, due to its integral behaviour, the response was too slow for sudden gust events. Third, a feed-forward controller was tested that uses single point wind speed measurements 1.44 D upstream of the rotor as a preview signal. A special signal processing algorithm was developed which enabled the controller to detect up-ramps early but also takes into account the slower advection time of down-ramps. As a result, the load overshoots in gust events have been significantly reduced which however comes with the price of increased

pitch actuation. The fourth controller version is a combination of the LFB and the FF controller.

     In conclusion, we want to outline four recommendations on controller design for very large wind turbines with a load constraint. First, model uncertainties need to be taken into account by a calibration or model update procedure. Second, a feedback from load measurements is crucial to fulfil the load constraint in spite of uncertainties. Third, a wind preview will be required

to reduce extreme loads on small time scales. And fourth, the TSR tracking needs to account for the actual pitch angle, which might be different than the expected pitch angle due to the advanced control methods that react on the turbine state and the environmental conditions.

     When we transfer the presented findings to the full-scale application, we need to evaluate the control strategies in the context

of very large rotors that are closely spaced in offshore wind farm applications. To achieve a more continuous and reliable supply with wind energy, we need large rotors to capture more energy in light winds. To fulfil the goals of installed capacity, those large rotors will need to be spaced relatively closely to each other and economic competitiveness can only be achieved by a lightweight and cost-effective turbine design. This can be supported by advanced control strategies that effectively reduce extreme loads during operation. The close spacing of the turbines will make highly turbulent partial wake scenarios more likely.

That means, spacially resolved information about the inflow needs to be provided to the controller and possibly individual pitch strategies will be needed to reduce unsymmetrical rotor loadings. A load feedback will be essential to correct for model uncertainties and to adapt the controller to changes in the system behaviour. Such changes could be present to the aerodynamic behaviour, e.g. due to leading edge erosion, or to the actuation and sensor system, e.g. due to long term drift.

Unconventional rotor concepts like the Hybrid-Lambda Rotor will need advanced control algorithms to unlock the full potential of aerodynamic efficiency and to ensure structural integrity. Wind tunnel experiments are a powerful tool to test the newly developed control strategies under tailored reproducible inflow conditions. With the results presented here, we have completed an important step in the validation of the Hybrid-Lambda rotor design and control methodology.



*Data availability.* The data shown in all figures of this paper is available under the following repository: https://doi.org/10.5281/zenodo.
16598898 (Ribnitzky, 2025)

## Appendix A

In this appendix, the scaling of the aerodynamic rotor torque $M_\mathrm{a}$ is explained, which is defined as:

$$M_\mathrm{a} = \frac{1}{2}\rho\pi R^3 u^2 c_\mathrm{q} \tag{A1}$$

Subject to scaling are the rotor radius $R$, the wind speed $u$ and the torque coefficient $c_\mathrm{q}$.

**Rotor radius:**

The scaling of the rotor radius can be expressed with

$$n_\mathrm{l} = \frac{R_\mathrm{m}}{R_\mathrm{f}} \tag{A2}$$

and this scaling criteria is satisfied.

**Time scaling:**

The time scaling is defined by the ratio of rotational speeds in the transition region:

$$n_\mathrm{t} = \frac{t_\mathrm{m}}{t_\mathrm{f}} = \frac{\omega_\mathrm{trans,f}}{\omega_\mathrm{trans,m}} \tag{A3}$$

and this scaling criteria is satisfied in the transition region.

**Torque coefficient:**

The torque coefficient can be derived from the power coefficient $c_\mathrm{p}$ and the TSR.

$$c_\mathrm{q} = \frac{c_\mathrm{p}}{\lambda} \tag{A4}$$

The torque and power coefficients are non-dimensional and they would be constant for an ideal scaling (identical aerodynamic characteristics and constant TSR). However, this is not the case for the given model turbine. The power coefficient is reduced due to aerodynamic losses associated with the lower Reynolds number. Thus, we introduce a scaling factor for the power coefficient:

$$n_\mathrm{cp} = \frac{c_\mathrm{p,m}}{c_\mathrm{p,f}} \tag{A5}$$

Further, the TSR is reduced in order to derive larger chord lengths and higher Reynolds numbers for the model turbine. We can define a scaling ratio for the TSRs in light-wind mode ($\lambda_\mathrm{LW}$) and strong-wind mode ($\lambda_\mathrm{SW}$). Since we address the maximum aerodynamic torque at rated wind speed, only the strong-wind mode is used here.

$$n_{\lambda,\mathrm{SW}} = \frac{\lambda_\mathrm{SW,m}}{\lambda_\mathrm{SW,f}} \tag{A6}$$





Consequently, we can define for the scaling of the torque coefficient:

$$n_{cq} = \frac{c_{q,m}}{c_{q,f}} = \frac{n_{cp}}{n_\lambda} \tag{A7}$$

**Wind speeds:**

For an ideal scaling (constant TSR), the wind speeds would scale with:

$$n_{u,\text{ideal}} = \frac{u_m}{u_f} = \frac{n_l}{n_t} \tag{A8}$$

Since we transferred the rotor design to lower TSRs, the wind speed scales with:

$$n_u = \frac{u_m}{u_f} = \frac{n_l}{n_t\, n_\lambda} \tag{A9}$$

The maximum rotational speed ($\omega_{\text{rated}}$) of the model turbine is constrained due to hardware limitations. When the maximum rotational speed is reached with the strong-wind TSR, this wind speed is considered as rated wind speed. Thus, the rated wind speed, rated torque and rated power change compared to the scaling theory. Or in other words, the model turbine is de-rated. This can be incorporated by defining a new time scaling for rated power:

$$n_{t,\text{rated}} = \frac{\omega_{\text{rated,f}}}{\omega_{\text{rated,m}}} \tag{A10}$$

The rated wind speed consequently scales with:

$$n_{u,\text{rated}} = \frac{u_{\text{rated,m}}}{u_{\text{rated,f}}} = \frac{n_l}{n_{t,\text{rated}}\, n_{\lambda,\text{SW}}} \tag{A11}$$

**Aerodynamic torque:**

If we combine equations A1, A2, A7 and A11 we can define a scaling ratio for the aerodynamic torque:

$$\boxed{n_{Ma} = n_l^3\, n_{u,\text{rated}}^2\, n_{cq,\text{rated}} = n_l^3 \left( \frac{n_l}{n_{t,\text{rated}}\, n_{\lambda,\text{SW}}} \right)^2 \frac{n_{cp,\text{rated}}}{n_{\lambda,\text{SW}}}} \tag{A12}$$

For the ideal scaling (constant TSR and constant $c_p$), equation A12 simplifies to:

$$n_{Ma}^* = \frac{n_l^5}{n_t^2} \tag{A13}$$

And if the time scaling equals the geometric scaling:

$$n_{Ma}^{**} = n_l^3 \tag{A14}$$

For the given model turbine, the required parameters are given in Table A1 and the coefficients from Eq. A12 result in:

$$n_{u,\text{rated}}^2\, n_{cq,\text{rated}} = 1.14 \tag{A15}$$

This is why the maximum aerodynamic torque of the model turbine is 14% higher, compared to the ideal and simplified scaling, where only the change in radius is considered.

These scaling laws can be extended from the aerodynamic torque to the generator torque, considering the different mechanical losses in the drive train, which we exclude here for the sake of brevity.



**Table A1.** Scaling parameters for the Hybrid-Lambda Rotor, relevant for the scaling of the aerodynamic torque.

| Parameter | Symbol | Model scale (subscript m) | Full-scale (subscript f) | Unit | scaling coefficient |
|---|---|---|---|---|---|
| Rotor diameter | $D$ | 1.8 | 326 | m | $n_l$ |
| Power coefficient at rated wind speed | $c_{p,rated}$ | 0.2805 | 0.2923 | - | $n_{cp,rated}$ |
| Maximum rotor speed | $\omega_{rated}$ | 577.3317 | 5.3824 | rpm | $1/n_{t,rated}$ |
| Rated wind speed | $u_{rated}$ | 9.0687 | 10.2 | m s$^{-1}$ | $n_{u,rated}$ |
| TSR at rated (SW mode) | $\lambda_{SW}$ | 6 | 9 | - | $n_{\lambda,SW}$ |
| Max. aerodynamic torque | $M_{a,max}$ | 5.2831 | $2.7563 \times 10^7$ | Nm | $n_{Ma}$ |

**Table A2.** Data necessary for the calculation of the time constants in Eq. 6.

| Parameter | Symbol | Model scale (subscript m) | Full-scale (subscript f) | Unit |
|---|---|---|---|---|
| Wind speed at the start of transition region | $u_{ts}$ | 6.3 | 6.8 | m s$^{-1}$ |
| Wind speed at the end of transition region | $u_{te}$ | 7.9 | 8.3 | m s$^{-1}$ |
| Derivative of aerodynamic torque coefficient at $u_{ts}$ | $\frac{dc_{q,ts}}{d\lambda}$ | $-7.6 \times 10^{-3}$ | $-3.7 \times 10^{-3}$ | - |
| Derivative of aerodynamic torque coefficient at $u_{te}$ | $\frac{dc_{q,te}}{d\lambda}$ | $-6.6 \times 10^{-3}$ | $-6.9 \times 10^{-4}$ | - |
| Derivative of generator torque at $u_{ts}$ | $\frac{dM_{g,ts}}{d\omega}$ | 0.1 | $5.9 \times 10^7$ | Nm s rad$^{-1}$ |
| Derivative of generator torque at $u_{te}$ | $\frac{dM_{g,te}}{d\omega}$ | 0.06 | $4.8 \times 10^7$ | Nm s rad$^{-1}$ |
| Air density | $\rho$ | 1.2 | 1.2 | kg m$^{-3}$ |

*Author contributions.* DR developed the control methodology for the Hybrid-Lambda Rotor; realized the controller versions on the real-time platform for the model turbine; planned, conducted and post-processed the experiments and wrote the paper. VP initiated the torque control concept for the Hybrid-Lambda Rotor; supervised the controller development and assisted in the controller tuning during the wind tunnel experiments. MK contributed to the control methodology with supporting discussions and supervised the investigations. All co-authors thoroughly reviewed the paper.

*Competing interests.* The contact author has declared that none of the authors has any competing interests.

*Acknowledgements.* The work presented in this paper was funded by the Deutsche Forschungsgemeinschaft (DFG, German Research Foundation) - Project-ID 434502799 - SFB 1463.



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
