# Peer review of "Experimental investigation of wind turbine controllers for the Hybrid-Lambda Rotor"

_Wind Energy Science, 2025_

## Author Comment (AC1)

Dear Paul Flemming,

We have the pleasure of submitting our revised paper "Experimental investigation of wind
turbine controllers for the Hybrid-Lambda Rotor" (wes-2025-143) for consideration in the
journal Wind Energy Science.

We are very grateful for the constructive feedback with lots of valuable suggestions from
the editorial team and the reviewers which helped to improve our paper. In short, we
want to highlight the major changes and additions:

• We included more concise takeaway messages throughout the paper, especially in
the figure captions.
• We clarified the red thread and the motivation which originates from very large
offshore turbines and we supported the statements in the introduction with
further citations.
• We restructured the methodology section so that the descriptions of all controller
sub-models are listed consecutively.
• We discussed our research in relation to further studies in literature, especially
comparing the methods to the work from Pusch et al. (2024) and Lazzarini et al.
(2025).
• We summarized the benefits and drawbacks of each controller in a table covering
all test cases which facilitates the readers to draw clear conclusions.
• We listed all symbols and abbreviations in a table in the appendix.
• We updated all figures to vector graphics and improved the colour schemes.

Furthermore, we have made all the necessary requested changes and have addressed all
comments of the reviewers (printed in black) in the detailed response below.

Our responses to the referees are written in green.

Reformulated or added phrases for the revised manuscript are cited with blue fonts.

Line, figure and table numbers in our answers are according to the revised manuscript.
Line, figure and table numbers in the referees' comments are according to the initial
manuscript. All updated figures are appended to this authors' response.

We feel that based on the reviewers comments our paper has been sharpened and
improved, especially in terms of clarity, presentation quality and additional
considerations. If any responses are unclear, or if you wish for additional changes, please
let us know.

Sincerely,

Daniel Ribnitzky

- On behalf of all authors –

**Referee 1:**

**### General comments**

This works presents the result of an experimental wind tunnel study on the scaled hybrid-lambda rotor using four different control strategies. The amount of tests performed and the quality of the experiments is high and I would like to thank the authors for their efforts. I also think the scientific significance of this work is very good as the hybrid-lambda rotor concept and the different controller strategies here push the boundaries of what is currently available in academic literature and the authors make good recommendations for large-scale wind turbine controls from their work. Overall, I think this work is suitable for wind energy science and should be accepted after minor revisions.

We thank the referee for the overall positive feedback and the very timely submission.

I recommend the following main improvement for the manuscript:

1. The objective of the work is not aligned well between the abstract, introduction, and the conclusion, which becomes apparent when first reading the abstract, introduction, and then the conclusion before continuing with the rest of the paper. From reading the abstract the goal is to develop and apply control strategies on the MoWiTO and this objective is repeated in the introduction. However, the introduction doesn't include what type of controllers will be tested which makes it unclear to the reader what exactly the 3rd paragraph in the introduction is for (it doesn't connect that well to the rest of the introduction). Finally, in the conclusion emphasis is given how this experiment leads to recommendations for controller design for very large wind turbines. I think announcing this as an objective in the introduction would make it come less unexpectedly. I think most of this can be solved by rewriting the introduction. I recommend elaborating on the objective and connecting it to the earlier parts in the introduction.

We started the paragraph which explains the paper objectives (lines 64 ff.) with "...to develop control methodologies for very large wind turbines with a load constraint". With this introductory sentence we want to remind the reader that our motivation originates from the challenge of very large offshore wind turbines. But we evaluate our approach on scaled wind turbines in the wind tunnel. To close the loop, we draw conclusions from our scaled experiments and we derive recommendations for the full-scale application. To highlight this red thread, we numbered the paper objectives:

The experimental testing aims to answers the following six specific research aspects. First, we address how the transition between the operating modes can be accomplished. Second, we investigate how we can achieve TSR tracking while the pitch controller is simultaneously active to constrain the loads. Third, we explore different methods on how to constrain the extreme operating loads. Fourth, we discuss how to deal with model uncertainties. Fifth, we evaluate the controller versions considering extreme loads, fatigue, pitch actuation and power output. And sixth, we transfer the findings from the scaled experiments to the full-scale application and derive recommendations for
controller design for very large offshore wind turbines with a load constraint.

The 3rd paragraph in the introduction is meant as a literature review and a state-of-the-
art report. To clarify this, we added:

Limiting extreme operating loads is often addressed by conventional peak-shaving and
this paragraph serves as a brief literature review on controller design which addresses
load constraints. For conventional peak-shaving …

A few smaller comments also apply more generally to the manuscript and are:

1. The resolution of the figures is quite small making it not pleasant to zoom in to see
more detail as a reader. I would recommend exporting figures using a vector format.

We increased the resolution for all figures and exported the figures to vector graphics
wherever technically possible. Unfortunately, vector graphics cannot contain half
transparent areas. That's why Fig. 10 and 14 are saved as jpg. Nevertheless, we ensured
that the resolution is sufficient.

2. The colour choice in figures is not always good. In black-and-white printing some
colours are almost indistinguishable, such as baseline and FFLFB in figure 13. I also
suspect that readers with a colour-vision deficiency would have difficulty distinguishing
the different colours in most figures. I would recommend using scientific colour maps, as
recommended by WES: https://zenodo.org/record/1243862

This was also addressed by the editorial team and we updated the figures 9, 10, 12, and
14 to ensure that readers with colour vision deficiencies can correctly interpret the
figures. More precisely, we used dashed lines for the FFLFB controller to improve the
distinctiveness to the baseline controller.

3. In the text, some lines in the figures are referred to by their colour instead of by their
label. I think that this exacerbates the problem with the colour choice in the figures.

We used colour names to describe different lines in Figures 2 and 7.

For Fig. 2, we replaced "blue" with "blue dashed-dotted" and "red" with "red dotted". We
further added labels to the red and blue shaded areas (A1, A2 and A3) to allow readers
with a colour-vision deficiency to correctly interpret Figure 2.

For Fig. 7 we added the labels in addition to the colour names to the text:

red line, labelled "filtered" in Fig. 7a yellow line, labelled "fixed propagation" in Fig. 7a yellow line, labelled "output, final propagation" in Fig. 7b

4. Not all lines in the figures are present in the legend and are sometimes only explained
in the caption. I think that afraid this exacerbates the problem with the colour choice in
the figures.

We thoroughly checked all figures for incomplete legends. For Fig. 10, 12 and 14 we
included all horizontal black dashed lines in the legends, or we included a label next to
the line.

The updated figures can be found in the appendix of this author response.

5. The captions of the figures currently have to explain more what is in the figures (also
due to incomplete legends) so to understand the results I have to switch from the figure
to the text and back a lot. It would improve readability if the captions would also include
the main result to make the figures more interpretable on their own.

We added additional descriptions and take-away messages to the figure captions where
we found it appropriate:

Fig. 7: The final output signal detects up-ramps early but delays down-ramps.

Fig. 9: The baseline controller works well in following the TSR of 6 in the SW mode, but the
loads are higher than expected due to model uncertainties. The LFB controller uses higher
pitch angles and is able to correctly constrain the loads. However, this leads to reduced
aerodynamic torque and consequently lower rotational speed and lower TSRs in the SW
mode. Further the power output of the LFB controller is lower in the SW mode compared
to the baseline controller.

Fig. 10: The LFB and FFLFB controllers can limit the loads in steady state perfectly to the
load constraint. During the gust, the baseline and LFB controllers react too slow and show
the highest load overshoots. The FF and FFLFB controllers react in advance and reduce
the load overshoot. The FF controller further reduces the variations in rotational speed
and torque.

Fig. 12: All controllers are able to follow the LW-TSR of 7.5 and the torque controller can
maintain the rotational speed constant in the transition region. In the SW mode, the LFB
controller uses higher pitch angles, effectively constraining the loads but resulting in TSRs
that are lower than the set-point of 6.

Fig. 13: Only for very large wake centre displacements, the single-point hot-wire
measurements converge towards the rotor effective wind speed which is predicted by the
wind speed estimator.

Fig. 14: The FF controller leads to the highest load overshoots for very large wake centre
displacements. The baseline controller estimates the rotor averaged wind speed much
better and shows lower loads, than the FF controller. The LFB and FFLFB controllers
perform best in limiting the loads.

Fig. 15: For the gust test case, the FF controller leads to around three times higher pitch
actuation, but reduces the fatigue loads. Also for the turbulent wind field test case, the FF
controller shows the highest pitch actuation, although the actuation is lower than for the
gust test case. The fatigue loads for the waked test case are strongly increased for all
controllers due to the non-uniformity of the inflow.

6. LUT1 and LUT2 are not very descriptive names for look-up tables, which made it slightly
harder to follow some parts of the text. I think a more descriptive naming would improved
the flow of the text.

Thank you for this suggestion. We substituted:

LUT1 → torque-LUT

LUT2 → pitch-LUT

**### Specific comments**

**1. Abstract:**

1. Line 2: "extreme loads" is usually associated with ultimate loads while it is also a trade-
off with fatigue loads. I think either removing "extreme" or explicitly adding fatigue loads
would be more complete.

We removed "extreme":

… master the trade-off between limiting the loads and maximizing power output …

**2. Introduction:**

1. Line 17: The first paragraph would benefit from one or two citations.

We added citations to support the statements in question:

The size of future wind turbine rotors is continuously increasing and the current trend of
growth in rotor diameter does not seem to saturate (Lüers, 2024; Berkhout et al., 2018;
Hand et al., 2018). Rotors with a low-specific rating can capture relatively more energy in
light winds which is beneficial for a reliable and more continuous electricity supply and it
will increase the value of wind energy, as stated among others by Hirth and Müller (2016);
Johnson et al. (2021); Wiser et al. (2021).

Lüers, S.: Status of Offshore Wind Energy Development in Germany Year 2024, Deutsche
Windguard, https://www.windguard.de/jahr-2024.html (last access: 3rd November 2025),
2024.

Berkhout, V., et al.: Windenergie Report Deutschland 2018, Fraunhofer-Gesellschaft,
https://doi.org/10.24406/publica-fhg-299720 (last access: 3rd November 2025), 2018.

Hand, M., et al.: IEA Wind TCP Task 26 – Wind Technology, Cost, and Performance Trends in Denmark, Germany, Ireland, Norway, Sweden, the European Union, and the United States: 2008–2016, National Renewable Energy Laboratory, NREL/TP-6A20-71844, Golden (US), https://doi.org/10.2172/1525772, 2018.

Hirth, L. and Müller, S.: System-friendly wind power, Energy Economics, 56, 51–63, https://doi.org/10.1016/j.eneco.2016.02.016, 2016.

Johnson, N., Paquette, J., Bortolotti, P., Mendoza, N., Bolinger, M., Camarena, E., Anderson, E., and Ennis, B.: Big Adaptive Rotor Phase I Final Report, National Renewable Energy Laboratory, NREL/TP-5000-79855, https://doi.org/10.2172/1835259, 2021.

Wiser, R., Millstein, D., Bolinger, M., Jeong, S., and Mills, A.: The hidden value of large-rotor, tall-tower wind turbines in the United States, Wind Engineering, 45, 857–871, https://doi.org/10.1177/0309524X20933949, 2021.

2. Line 18: "Rotors with a low-specific rating can capture more energy in light winds" is technically not correct. I think adding making it "relatively more energy" would read better.

We added "relatively":

Rotors with a low-specific rating can capture relatively more energy in light winds...

3. Line 30: Reading this paragraph, I feel like I wouldn't know at which paper to start for a good general overview of the Hybrid-Lambda concept because the two citations used are introduced rather late as simulation studies of the concept (not as fundamental papers explaining the concept). Maybe it's good to cite one of your works earlier in the paragraph so that readers have a clear citation which they can learn about the concept of Hybrid-Lambda.

We added a citation at the beginning of the paragraph:

This problem is addressed by the Hybrid-Lambda rotor design methodology, as first described by Ribnitzky et al. (2024).

**3. Methodology**

1. The order of subsections and subsubsections is first an overview of the steady-state operating points, then the scaling considerations, then the torque controller, and then the pitch controller. After finishing the section on steady-state operating points I was expecting more information on the implementation of the controllers, as you already hint to some of the challenges that the hybrid-lambda concept poses for control. I would put the scaling considerations section somewhere else in the methodology so the control parts can be together.

We placed the scaling considerations after the subsections about the torque and pitch controllers. We adjusted the introductory paragraph of the methods section accordingly:

In this section, we first explain the Hybrid-Lambda control methodology. We then address
the torque controller in Sect. 2.1.1 and we outline four different versions of the pitch
controller in Sect. 2.1.2. The transferability of the experimental results from wind tunnel
scale to the full-scale turbine is demonstrated in Sect. 2.1.3. The experimental set-up ...

2. Line 171-178: This is a good summary of the scaling considerations which makes this
section possible to follow for people with less exposure to wind tunnel testing.

Thank you for this positive comment.

3. Line 201-216: I understand this scheme and think it's an interesting way to solve this
problem but I had to reread this section a couple of times to convince myself that it works.
I would recommend to rewrite this section and take the reader along in the process to
make it more intuitive and understandable.

We added introductory sentences to each paragraph which explain the task of the torque
controller for the described region a priori to facilitate the understanding:

If the rotational speed is between $\omega_{min}$ and $\omega_{middle,LW}$, the task of the torque controller
is either to maintain $\omega_{min}$ or to track the light-wind TSR.

If the rotational speed is between $\omega_{middle,LW}$ and $\omega_{middle,SW}$, the task of the torque
controller is to track the light-wind TSR, to maintain $\omega_{trans}$, or to track the strong-wind
TSR.

4. Line 222: "Additionally, we tested a combination of the latter two versions (FFLFB)." I
would combine this with the previous sentence because as a reader I expected 4
controllers to be listed in one sentence.

We adjusted the wording accordingly:

We introduce four versions of the pitch controller - a baseline controller, a load feedback
(LFB) controller, a feed-forward (FF) controller and a combination of the latter two versions
(FFLFB).

5. Line 223-224: I think this is a creative and interesting approach. I would add a short
explanation as to why this works. I guess the reasoning is that below the maximum rotor
speed the PI controller always wants to speed up thus decrease the pitch but is then
saturated by the minimum pitch angle.

We added an explanation at the end of the paragraph:

With this architecture, the controller always wants to increase the rotational speed in the
partial load range, thus reducing the pitch angle. But it is saturated by the minimum pitch
angle which realizes the load constraint.

**4. Discussion**

1. I'm familiar with the work of Lazzerini et al (2025) and it would be helpful to see a comparison to your method. I see their work as completely letting go of the concept of tip speed ratio, or going towards a kind of Hybrid-Lambda control concept with an infinite number of TSRs instead of 2. Do you think that their control approach would work well for your rotor design?

2. In extension to that, their torque controllers also use the wind speed estimate. Do you think that that would have worked in your case too and I'm curious why you made the decision to not use the wind speed estimate in your torque controller.

Prior to conducting this study, we applied the framework from Pusch et al. (2024) on the Hybrid-Lambda rotor. This resulted in an alternative operating strategy where the rotational speed is kept constant for $u > u_{ts}$, not following the strong-wind TSR at all. However, the improvements in the power output were rather small (1% increase in AEP for the full-scale 15 MW Hybrid-Lambda turbine). Further, this strategy comes along with other disadvantages such as an increased rated generator torque and reduced stall margins for the operational angle of attack. Therefore, we decided not to include the alternative strategy in this paper.

Unfortunately, the work of Lazzerini et al. (2025) was submitted after the experiments presented in this paper were already finished (in August 2024). We see a lot of potential in applying the framework of Lazzerini et al. (2025) on the Hybrid-Lambda rotor.

An advantage would be that the wind speed estimate is also used for the torque controller. That means, for every wind speed there is a unique assignment of a generator torque. Whereas for our approach, the input is the rotational speed and for $\omega_{trans}$ multiple values of generator torque are assigned. This is where the PI-controller is needed.

Since Lazzerini et al. (2025) use a PI torque controller to track the rotational speed set-points, the controller should perform well also if an additional load feedback changes the pitch angle.

However, Lazzerini et al. (2025) realize the optimized operating points with a controller framework that is essentially based on a reliable wind speed estimator. The wind speed estimate is used for the pitch and torque controller as well as to derive the rotational speed set-point. This makes the approach very vulnerable to model uncertainties which could be present in the wind speed estimator and in the LUTs which contain the optimized steady-state operating points. We expect a load feedback to be a necessary addition to the framework proposed by Lazzerini et al. (2025), when implemented on the Hybrid-Lambda rotor.

We decided to add a condensed version of these considerations to the discussion chapter:

In this study, we derived the steady-state operating points by prescribing a schedule of rotational speed which consists of tracking the light-wind and strong-wind TSR and a transition region with constant rotational speed in-between. The pitch angle was then derived by optimizing the power coefficient with respect to the load constraint. Here, we want to emphasise that also other approaches are possible. Frameworks for the steady-state operating points, as described for example by Pusch et al. (2024) and Lazzerini et al. (2025), enable the optimization of both the rotational speed and the pitch angle for the entire operating range. The control philosophy described by Lazzerini et al. (2025) drops the assumption of constant TSR or it could be interpreted as a control strategy with an infinite number of TSRs. This approach could be promising for the Hybrid-Lambda concept and could be implemented in future studies. However, special care needs to be taken considering model uncertainties because the control algorithm proposed by Lazzerini et al. (2025) is essentially based on a reliable wind speed estimator. Possibly, a load feedback would be a necessary addition.

**5. Conclusions**

1. Line 558: "arranging" is too vague in this context and makes it hard to understand for a reader who has only read the abstract and introduction.

We reformulated with a more descriptive summary. However, we would like to avoid lengthy repetitions in the conclusions:

We showed how two different TSRs can be tracked below rated wind speed by using three different rotor speed set points ($\omega_{min}$, $\omega_{trans}$ and $\omega_{max}$) and by defining the generator torque saturations according to the TSRs. With this method, we can further realize a transition region with constant rotational speed ($\omega_{trans}$) to switch between the two operational TSRs.

2. The first two paragraphs of the conclusion are clear. However, the third and fourth paragraph discuss the transfer from the presented control strategy to a full-scale system and make a concluding remark of the work. I thought that the full-scale discussion seemed to stand on its own rather than be explicitly related to this study. The last paragraph could be made clearer by defining a more specific point. I would revise these sections by defining a main point you want to make and making the connection to your work clearer.

With the third paragraph we want to close the loop in the storyline and connect back to the introduction where we introduced the need for low-specific-rating turbines which produce more power in light winds and therefore increase the value of wind energy. This is the overarching motivation, namely cannibalization of wind energy, the need for larger rotors and low-specific ratings, lightweight and cost-effective blade design and consequently the need for advanced control strategies that reduce the extreme loads. We tried to emphasis this red thread with a couple of keywords:

To close the loop, we transfer the presented findings to the full-scale application. We need to evaluate the control strategies in the context of very large rotors that are closely spaced in offshore wind farm applications. As explained in the introduction, larger rotors are needed to generate more energy in light winds and thus achieve a more continuous and reliable supply of wind energy...

**Technical corrections**

I have no technical corrections to recommend.

**Referee 2:**

**Overall**

- This is an interesting investigation of a few control schemes in an experimental setting. The control schemes are scaled and based on the Hybrid Lambda rotor. The authors have put a lot of thought and effort into the scaling of the controller, but it's not clear how the algorithms will perform or need to be changed at full-scale. The results are thoroughly presented, but clear takeaways and the impact of this work are unclear.

We thank the referee for the feedback. We agree that a significant contribution of our work lies in the methodology of how wind turbine controls can be scaled to wind tunnel size. We thoroughly explain which metrics need to be checked in order to allow a transfer of the findings from wind tunnel scale back to the full-scale application. This is a key point that we found missing in most of the previous works on scaled controller evaluations. To further emphasis the transferability, we added to the introduction section:

The experimental testing aims to answers the following six specific research aspects. (…) And sixth, we transfer the findings from the scaled experiments to the full-scale application and derive recommendations for controller design for very large offshore wind turbines with a load constraint.

We moved the recommendations for the full-scale application to the discussion section where we systematically evaluated the achievements of the six research aspects outlined in the introduction:

We first discuss the achievement of the six research questions outlined in the introduction. We give recommendations on controller design for very large wind turbines with a load constrain and we address challenges in the scaling approach. We conclude the discussion by comparing our finding to related research. (…)

The sixth research objective was to outline recommendations on controller design for very large wind turbines with a load constraint. First, model uncertainties need to be taken into account by a calibration or model update procedure. Second, a feedback from load measurements is crucial to fulfil the load constraint in spite of uncertainties. Third, a wind preview will be required to reduce extreme loads on small time scales. Although load limiting control concepts without a preview already exist on commercial turbines, the tremendous size of the turbines addressed in this study (diameter of more than 300 m) as well as the wide wind speed range in which the load constraint is active (upper half of the partial load range) will require more sophisticated techniques to achieve an economical viable design. And fourth, the TSR tracking needs to account for the actual pitch angle, which might be different than the expected pitch angle due to the advanced
control methods that react on the turbine state and the environmental conditions.

We essentially improved our paper by including more concise takeaway messages
throughout the paper, especially in the figure captions as reported in line 122-149 of this
author response letter.

**Major Comments**

• If the main contribution of this controller is to achieve a higher power coefficient
when the loads are constrained, this should be made clear early in the paper. Is
this outcome validated by the experiments?

In this context, the benefits of the Hybrid-Lambda rotor are twofold: First, the power
output is increased due to the tremendously enlarged rotor area especially in light winds
where the load constraint is not active. This will increase the value of wind energy and
counterbalances cannibalization of wind energy. Second, whenever the loads need to be
constrained, the Hybrid-Lambda rotor can operate at better power coefficients compared
to conventional rotors of the same size which realize the same load constraint with
conventional peak shaving. We provide evidence for this already in our previous
publication (Ribnitzky et al. 2024). This is only possible with a close connection of the blade
design and control strategy. Proving this statement (better power coefficient when the
loads are constrained) was not an objective of the presented study. This would have
required a comparison with a wind turbine model that represents a conventional full-scale
turbine using a conventional controller which was not part of the measurement
campaign. In this paper we concentrate on the different controller designs and on how
the above-mentioned benefits are maintained under dynamic operating conditions.

To clarify the objective, we added your suggestion to the first paragraph of the
introduction:

This emphasises the necessity of advanced control algorithms which limit the extreme
loads whenever needed but maximize the power output whenever possible. In other
words, a close interaction of blade design and controller design methodology is needed,
with the aim of achieving better power coefficients when the loads are constrained. This
problem is addressed by the Hybrid-Lambda rotor design methodology...

• It is good to scale the maximum pitch rate, but in practice this should not be
reached during normal operation. Have you scaled the bandwidth of the pitch
controller's response to a gust as well?

We investigated how the bandwidth of the pitch response compares to the gust excitation
and to the full-scale equivalent. We added the additional analysis to the results section of
the gust test case in Sect. 3.2. The new figure can be found in the appendix of the author
response letter.

To further investigate the speed at which the pitch controllers are active, we derived the
power spectral density of the pitch signal, and we compare it to the gust excitation in

Fig. 11. In this analysis, we further include the simulation results for the full-scale 15 MW Hybrid-Lambda turbine of the design load case 2.3 extreme operating gust with a duration of 10.5 sec according to IEC 61400-1 (2019), simulated with the aero-servo-elastic tool openFAST. Note, that the wind speed signal from the IEC load case is not turbulent and the inflow has not been identically scaled across all frequencies. For the model turbine, the FF controller shows higher magnitudes for frequencies between 2 Hz and 10 Hz, compared to the baseline controller which is in line with the faster response of the FF controller, as presented above. Next, we compare the bandwidth of the scaled and the full-scale system. Note that system identification has not been the goal of the performed experiments, meaning that the system has not been adequately excited on all frequencies. Nevertheless, the spectra of the pitch signals still give an insight into the speed of the respective systems. The pitch spectra of the model turbine drop below -3 dB at around 1.1 Hz, whereas the full-scale spectra drop at 0.1 Hz. Considering the time scaling of $n_t = 1/114$, the model turbine controllers are about 10 times slower than ideally scaled. Since the gusts are also eight times slower than ideally scaled, as explained in Sect. 2.1.3, we can conclude that the controllers of the wind tunnel model represent a reasonable representation of the full-scale equivalent.

- It is unclear if there is an impact of this work outside of a Hybrid-lambda rotor. Please elaborate.

We believe that our work has a major impact to the wind energy research community:

- Scaling considerations when testing wind turbine controllers in a wind tunnel (Sect. 2.1.3).
- Recommendations for controller design for very large wind turbines with a load constraint which overarches a wind speed regime of several meters per second, e.g. the upper half of the partial load range (line 581-589).
- Recommendations for the design of controllers that move away from the conventional concept of one distinct operational TSR in the partial load range, as among others proposed by Lazzerini et al. (2025). (Sect. 2.1, 2.1.1 and 2.1.2)
- The disclosure of problems that arise if the pitch controller is active to constrain the loads while the torque controller simultaneously aims to set a desired rotational speed below rated power (lines 398-405, 471-474, 551-559, Fig. 9 and Fig. 12).
- A practical example of the impact of model uncertainties and how to overcome these hurdles with the use of a load feedback (lines 387-398, 566-574, Fig. 9 and Fig. 12).
- A methodology to process wind preview measurement data for the use in feed-forward controllers which aim to detect up-ramps early but delay down-ramps (lines 225-235, Fig. 6 and 7).

These contributions are explained in the last paragraph of the introduction which covers the objectives of the paper (line 64-74). Further, the achievements of the objectives are discussed in line 546-589.

- It is mentioned that at full scale, the root bending moment will have 1P oscillations, but at model scale, the 1 Hz filter on the RBM will remove this frequency. It seems like a different filtering method will be needed at larger scales. What other
considerations in the controller must be made when scaling up?

We agree that the filter for the RBM measurements should be improved in future studies.
Due to the scaling, a high rotational speed is necessary for the model turbine and
associated vibrations increase the noise level on the strain gauge measurements. Of
course, for a full-scale application, every controller parameter that is impacted by the time
scaling needs to be adjusted (filter, tuning parameter, hardware constraints, rate limiter,
etc.). A full-scale turbine operates at lower rotational speed, the structure is subject to
lower eigenfrequencies, and the RBM measurements will be of better quality. We added
a clarification to the description of the LFB controller:

Most likely, a commercial full-scale turbine will provide RBM measurements with better
signal to noise ratio and with a dedicated filter design, also 1P load variations can be
captured accurately.

• In Fig 10, the FF controllers have a higher pitch angle, but the same speed and
torque at the same wind speed. How can you account for this?

For the gust event itself (e.g. 2 s < t < 3.5 s): The FF controllers (FF and FFLFB) show higher
pitch angles during the gust event, at least compared to the baseline controller. But, the
variations in rotational speed and torque are reduced, compared to the baseline and LFB
controller which is already addressed in line 434-436.

For the steady-state period (e.g. 0 s < t < 2 s), the FF controller shows lower pitch angles,
compared to the LFB and FFLFB controller. In other words, the pitch angle is about 1.3°
lower for the FF and baseline controllers, compared to the LFB and FFLFB controllers. They
all show similar measured torque at the low-speed shaft and similar rotational speed. To
explain why different pitch angles don't lead to different rotational speeds (as it was seen
for the SW mode in Fig. 9a and as explained in lines 399 ff.), we analyse the torque
coefficient for these two operating modes.

Before the gust event (0 s < t < 2 s), the turbine is in LW mode at an expected TSR of 7.5
and pitch angles close to zero. For those pitch angles, the slope of the $c_q$-$\beta_{pitch}$-curve is
very flat. A difference in pitch angle between -0.3° and +1° leads to a difference in the
torque coefficient of 0.0026. This is equivalent to a difference in torque of 0.13 Nm, or
2.5% of the rated generator torque.

Whereas for the SW mode, the turbine operates at a TSR of 6 and at pitch angles above
+4°. For those higher pitch angles, the slope of the $c_q$-$\beta_{pitch}$-curve is much steeper. A
difference in pitch angle between +4° and +5.3° leads to a difference in the torque
coefficient of 0.0061. This is equivalent to a difference in torque of 0.54 Nm, or 10% of the
rated generator torque.

These calculations were performed using the $c_q$-$\lambda$-$\beta_{pitch}$ surface, derived from
measurement data from the rotor characterisation, as it is used for the wind speed
estimator.

Consequently, the impact of different pitch angles on the rotor speed is much more
prominent for the SW mode (Fig. 9a, 220 s < t < 280 s), compared to the LW mode (Fig. 10,
0 s < t < 2 s). We conclude that the presented measurements are in line with the
expectations from our model. We don't think that this detailed analysis will help to
improve the clarity of the paper and it would distract from the major take-aways.
Therefore, we only addressed it with a brief note at the beginning of Sect. 3.2 where the
different behaviours in steady-state are explained.

Note, that 1° difference in the pitch angle does not significantly change the rotor speed,
which is due to a low sensitivity of the aerodynamic torque to the pitch angle in that
region.

• One of the authors' conclusion is that a wind preview will be required? That's kind
of a strong statement given that load limiting control systems exist on modern
turbines without preview measurement.

Indeed, common load limiting control systems of commercial turbines do not use
upstream wind speed measurements. But the presented study goes beyond existing
commercial turbines. We are addressing turbines of more than 300 m diameter which are
tailored for an increased feed-in power in light winds, whereas the loads are constrained
for the upper half of the partial load range to foster a lightweight and cost-effective blade
design. Such turbines are not available on the market, neither do the required control
concepts exist. In Sect. 3.2 we showed how a preview can significantly reduce the extreme
loads and the performance without a feed-forward controller might not be sufficient for
innovative turbines of that size. We added an explanation to the conclusions:

Third, a wind preview will be required to reduce extreme loads on small time scales.
Although load limiting control concepts without a preview already exist on commercial
turbines, the tremendous size of the turbines addressed in this study (diameter of more
than 300 m) as well as the wide wind speed range in which the load constraint is active
(upper half of the partial load range) will require more sophisticated techniques to achieve
an economical viable design.

• While the results are presented in detail, a high-level summary would help the
reader navigate the article more easily. I'm not sure what to take way from these
investigations. And I'm not sure what the benefits and drawbacks of each
controller are.

We summarized the benefits and drawbacks of each controller in a table covering all test
cases and added to the conclusions chapter:

Four versions of the pitch controller, which aim to satisfy the constraint on the flapwise
RBM, were investigated. An overview of the findings is provided in Table 2. First, a baseline
controller...

**Table 2.** Advantages (+) and disadvantages (-) of all tested controller versions for all test cases.

| | Baseline | LFB | FF | FFLFB |
|---|---|---|---|---|
| **Wind steps** | + Good tracking of LW and SW TSR
(+) RBM constant in transition region and SW mode
- RBM constraint exceeded by 10% | + RBM correctly constrained
- TSR in SW mode too low
- Reduced power output | + Good tracking of LW and SW TSR
- RBM increasing with wind speed in transition region and SW mode
- RBM constraint exceeded by 10% | (not evaluated) |
| **Gusts** | - Highest load overshoot. RBM constraint exceeded by 35% | + Steady state RBM correct

- Slowest pitch reaction

- Load overshoot. RBM constraint exceeded by 26% | + Pitch is increased just before the gust arrives
+ Lowest load overshoot of 11%
+ Lowest variation in rotational speed
- Highest pitch actuation | + Steady state RBM correctly set
+ Longest period of increased pitch
+ Reduced load overshoot of 14% |
| **Turbulent inflow** | + Good tracking of LW and SW TSR
- Average RBM in transition and SW mode exceeds constraint by 6% | + RBM correctly constrained
- Reduced power output
- TSR in SW mode too low | - Average RBM in transition and SW mode exceeds constraint by 9% | (not evaluated) |
| **Waked inflow** | + Load overshoots are lower than for FF
- RBM constraint exceeded by 12% | + Lowest load overshoot, RBM constraint exceeded by 5% | - Highest load overshoots
- RBM constraint exceeded by 17%
- Power output is not increased
- Higher power variations | + Lowest load overshoot, RBM constraint exceeded by 6% |

**Minor Comments**

- The term "Control schedule" is unclear. Are these steady state operating points? Or are they inputs to the turbine? Which are inputs and which are outputs of the controller? Also, Fig 1. is far from where it's introduced and the various U_* are not defined

We replaced the term schedule with steady-state operating points throughout the manuscript.

In this section we describe the steady-state operating points for the Hybrid-Lambda Rotor, as depicted in Fig. 1 for the scaled wind tunnel model.

In contrast, this study focuses on how these steady-state operating points can be realised by a pitch and torque controller.

Compared to the steady-state operating points presented by Ribnitzky et al. (2025), the maximum rotational speed...

Figure 1. Steady-state operating points for the Hybrid-Lambda model turbine...

This data (generator torque over rotational speed) is acquired a priori...

We moved Fig. 1 closer to the first reference in the text in the revised manuscript. However, please note that the placement of the figures will be arranged by the editorial team of the journal for the final paper. Unfortunately, we as authors can not influence the position of the figures in the final published paper.

The variables $u_{ts}$ and $u_{te}$ are defined in lines 90 and 93, respectively. Additionally, they
are explained in the list of symbols. And we further added an explanation to the figure
caption:

$u_{ts}$ and $u_{te}$, wind speed at start and end of transition.

**Editorial Comments**

• In Line 24, what is "it?" Many sentences throughout this article start with it and this,
which are not always clear. Consider having this paper edited for grammar.

We use this grammar structure to split long sentences into two shorter ones to facilitate
easier reading. "It" or "this" refers to the last-mentioned subject. Wherever this was not
clear from the context we adjusted the grammar accordingly:

The baseline controller also includes a wind speed estimator, similar to that of a full-scale
turbine. ⤷ The estimator is based on filtered measurement signals…

The torque coefficient as a function of pitch angle and TSR was also derived from steady-
state measurements from previous wind tunnel campaigns. ⤷ The data was extended with
BEM simulations…

This results in a pitch increment ($\Delta\beta_{pitch}$) which can either be positive or negative, but it
is saturated with the maximum allowable pitch rate. ⤷ The pitch increment is then added…

• There are a lot of variables and acronyms defined throughout this text. It helps the
reader if they are defined in captions or in a table at the start of the article.

We listed all symbols and abbreviations in a table in the appendix and added a reference
to the beginning of the methodology section. We'll leave it to the editorial team whether
to place this table at the start of the article or in the appendix.

All symbols and abbreviations are listed in Table B1.

Appendix B: List of symbols and abbreviations… (not printed here in full, for sake of
brevity).

**Technical comments from the editorial board:**

1) Please add the ZIP code, city, and country to affiliation 1.

Carl von Ossietzky Universität Oldenburg, School of Mathematics and Science, Institute of Physics, 26129, Oldenburg, Germany

2) Please ensure that the colour schemes used in your maps and charts allow readers with colour vision deficiencies to correctly interpret your findings. Please check your figures using the Coblis – Color Blindness Simulator (https://www.color-blindness.com/coblis-color-blindness-simulator/) and revise the colour schemes accordingly. => Figs. 9, 10, 11, and 13.

We updated the figures 9, 10, 12, and 14 to ensure that readers with colour vision deficiencies can correctly interpret the figures.

**Appendix: Updated and new figures:**

Full resolution can be found in the revised manuscript.

[Figure]

**Figure 1.** Steady-state operating points for the Hybrid-Lambda model turbine, derived from steady-state measurement data. Background colours indicate the operating mode: green, light-wind mode (LW); yellow, transition (TS); red, strong-wind mode (SW). $u_{ts}$ and $u_{te}$, wind speed at start and end of transition.

[Figure]

**Figure 2.** (a) Rotational speed and generator torque as a function of wind speed, derived from BEM simulations. (b) Saturations and set points for the torque controller. Shaded areas indicate the permissible generator torque for a given rotational speed.

[Figure]

**Figure 9.** Turbine response with the baseline, LFB and FF controllers to the test case with wind steps: Time series (a) and statistical analysis of RBM (b) and power (c). Background colours indicate the operating mode: green, light-wind mode (LW); yellow, transition (TS); red, strong-wind mode (SW). Middle markers, median; boxes, 25th and 75th percentiles; whiskers, minimum and maximum of the considered data. The baseline controller works well in following the TSR of 6 in the SW mode, but the loads are higher than expected due to model uncertainties. The LFB controller uses higher pitch angles and is able to correctly constrain the loads. However, this leads to reduced aerodynamic torque and consequently lower rotational speed and lower TSRs in the SW mode. Further the power output of the LFB controller is lower in the SW mode compared to the baseline controller.

[Figure]

**Figure 10.** Turbine response to gust events. Solid lines show the ensemble average over 50 gust repetitions. The shaded areas indicate the 95% confidence intervals. Wind speed measurements are performed 2.7 m upstream of the rotor and are not propagated in this plot. The LFB and FFLFB controllers can limit the loads in steady state perfectly to the load constraint. During the gust, the baseline and LFB controllers react too slow and show the highest load overshoots. The FF and FFLFB controllers react in advance and reduce the load overshoot. The FF controller further reduces the variations in rotational speed and torque.

[Figure]

**Figure 11.** Power spectral density of **(a)** the pitch signal and **(b)** the wind speed excitation for the gust test case in the wind tunnel and for aero-servo-elastic simulations with extreme operating gust for the 15 MW full-scale (FS) Hybrid-Lambda turbine. The model turbine (MT) controllers are about 10 times slower than ideally scaled ($n_t = 1/114$). This corresponds to the gust duration which is eight times slower than ideally scaled.

[Figure]

**Figure 12.** Turbine response to turbulent inflow. Background colours indicate the desired operating modes: Green, light-wind (LW); red, strong-wind (SW); yellow, transitioning between the operating modes. All controllers are able to follow the LW-TSR of 7.5 and the torque controller can maintain the rotational speed constant in the transition region. In the SW mode, the LFB controller uses higher pitch angles, effectively constraining the loads but resulting in TSRs that are lower than the set point of 6.

[Figure]

**Figure 14.** Turbine response to waked inflow. **(a)** $200\,s$ and **(b)** $60\,s$ excerpt of the full test case. Background colours indicate the desired operating modes: green, light-wind; yellow, transitioning between the operating modes. The FF controller leads to the highest load overshoots for very large wake centre displacements. The baseline controller estimates the rotor averaged wind speed much better and shows lower loads, than the FF controller. The LFB and FFLFB controllers perform best in limiting the loads.